# Prediction, Consistency, Curvature: Representation Learning for Locally-Linear Control

**Nir Levine**[1]*, **Yinlam Chow**[2]*, **Rui Shu**[3], **Ang Li**[1], **Mohammad Ghavamzadeh**[4], **Hung Bui**[5]
[1]DeepMind, [2]Google Research, [3]Stanford University, [4]Facebook AI Research, [5]VinAI

## ABSTRACT

Many real-world sequential decision-making problems can be formulated as optimal control with high-dimensional observations and unknown dynamics. A promising approach is to embed the high-dimensional observations into a lower-dimensional latent representation space, estimate the latent dynamics model, then utilize this model for control in the latent space. An important open question is how to learn a representation that is amenable to existing control algorithms? In this paper, we focus on learning representations for locally-linear control algorithms, such as iterative LQR (iLQR). By formulating and analyzing the representation learning problem from an optimal control perspective, we establish three underlying principles that the learned representation should comprise: **1)** accurate prediction in the observation space, **2)** consistency between latent and observation space dynamics, and **3)** low curvature in the latent space transitions. These principles naturally correspond to a loss function that consists of three terms: *prediction*, *consistency*, and *curvature* (PCC). Crucially, to make PCC tractable, we derive an amortized variational bound for the PCC loss function. Extensive experiments on benchmark domains demonstrate that the new variational-PCC learning algorithm benefits from significantly more stable and reproducible training, and leads to superior control performance. Further ablation studies give support to the importance of all three PCC components for learning a good latent space for control.

## 1 INTRODUCTION

Decomposing the problem of decision-making in an unknown environment into estimating dynamics followed by planning provides a powerful framework for building intelligent agents. This decomposition confers several notable benefits. First, it enables the handling of sparse-reward environments by leveraging the dense signal of dynamics prediction. Second, once a dynamics model is learned, it can be shared across multiple tasks within the same environment. While the merits of this decomposition have been demonstrated in low-dimensional environments (Deisenroth & Rasmussen, 2011; Gal et al., 2016), scaling these methods to high-dimensional environments remains an open challenge.

The recent advancements in generative models have enabled the successful dynamics estimation of high-dimensional decision processes (Watter et al., 2015; Ha & Schmidhuber, 2018; Kurutach et al., 2018). This procedure of learning dynamics can then be used in conjunction with a plethora of decision-making techniques, ranging from optimal control to reinforcement learning (RL) (Watter et al., 2015; Banijamali et al., 2018; Finn et al., 2016; Chua et al., 2018; Ha & Schmidhuber, 2018; Kaiser et al., 2019; Hafner et al., 2018; Zhang et al., 2019). One particularly promising line of work in this area focuses on learning the dynamics and conducting control in a low-dimensional latent embedding of the observation space, where the embedding itself is learned through this process (Watter et al., 2015; Banijamali et al., 2018; Hafner et al., 2018; Zhang et al., 2019). We refer to this approach as learning controllable embedding (LCE). There have been two main approaches to this problem: **1)** to start by defining a cost function in the high-dimensional observation space and learn the embedding space, its dynamics, and reward function, by interacting with the environment in a RL fashion (Hafner et al., 2018; Zhang et al., 2019), and **2)** to first learn the embedding space and its dynamics, and then define a cost function in this low-dimensional space and conduct the control (Watter et al., 2015; Banijamali et al., 2018). This can be later combined with RL for extra fine-tuning of the model and control.

In this paper, we take the second approach and particularly focus on the important question of what desirable traits should the latent embedding exhibit for it to be amenable to a specific class of control/learning algorithms, namely the widely used class of locally-linear control (LLC) algorithms? We argue from an optimal control standpoint that our latent space should exhibit three properties. The first is prediction: given the ability to encode to and decode from the latent space, we expect

---

*Equal contribution. Correspondence to `nirlevine@google.com`

the process of encoding, transitioning via the latent dynamics, and then decoding, to adhere to the true observation dynamics. The second is consistency: given the ability to encode a observation trajectory sampled from the true environment, we expect the latent dynamics to be consistent with the encoded trajectory. Finally, curvature: in order to learn a latent space that is specifically amenable to LLC algorithms, we expect the (learned) latent dynamics to exhibit low curvature in order to minimize the approximation error of its first-order Taylor expansion employed by LLC algorithms. Our contributions are thus as follows: (1) We propose the Prediction, Consistency, and Curvature (PCC) framework for learning a latent space that is amenable to LLC algorithms and show that the elements of PCC arise systematically from bounding the suboptimality of the solution of the LLC algorithm in the latent space. (2) We design a latent variable model that adheres to the PCC framework and derive a tractable variational bound for training the model. (3) To the best of our knowledge, our proposed curvature loss for the transition dynamics (in the latent space) is novel. We also propose a direct amortization of the Jacobian calculation in the curvature loss to help training with curvature loss more efficiently. (4) Through extensive experimental comparison, we show that the PCC model consistently outperforms E2C (Watter et al., 2015) and RCE (Banijamali et al., 2018) on a number of control-from-images tasks, and verify via ablation, the importance of regularizing the model to have consistency and low-curvature.

## 2 PROBLEM FORMULATION

We are interested in controlling the non-linear dynamical systems of the form $s_{t+1} = f_{\mathcal{S}}(s_t, u_t) + w$, over the horizon $T$. In this definition, $s_t \in \mathcal{S} \subseteq \mathbb{R}^{n_s}$ and $u_t \in \mathcal{U} \subseteq \mathbb{R}^{n_u}$ are the state and action of the system at time step $t \in \{0, \dots, T-1\}$, $w$ is the Gaussian system noise, and $f_{\mathcal{S}}$ is a smooth non-linear system dynamics. We are particularly interested in the scenario in which we only have access to the high-dimensional observation $x_t \in \mathcal{X} \subseteq \mathbb{R}^{n_x}$ of each state $s_t$ ($n_x \gg n_s$). This scenario has application in many real-world problems, such as visual-servoing (Espiau et al., 1992), in which we only observe high-dimensional images of the environment and not its underlying state. We further assume that the high-dimensional observations $x$ have been selected such that for any arbitrary control sequence $U = \{u_t\}_{t=0}^{T-1}$, the observation sequence $\{x_t\}_{t=0}^{T}$ is generated by a stationary *Markov* process, i.e., $x_{t+1} \sim P(\cdot|x_t, u_t), \ \forall t \in \{0, \dots, T-1\}$.[1]

A common approach to control the above dynamical system is to solve the following stochastic optimal control (SOC) problem (Shapiro et al., 2009) that minimizes expected cumulative cost:

$$\min_{U} \quad L(U, P, c, x_0) := \mathbb{E}\Big[c_T(x_T) + \sum_{t=0}^{T-1} c_t(x_t, u_t) \mid P, x_0\Big],\text{[2]} \tag{SOC1}$$

where $c_t : \mathcal{X} \times \mathcal{U} \to \mathbb{R}_{\geq 0}$ is the immediate cost function at time $t$, $c_T \in \mathbb{R}_{\geq 0}$ is the terminal cost, and $x_0$ is the observation at the initial state $s_0$. Note that all immediate costs are defined in the observation space $\mathcal{X}$, and are bounded by $c_{\max} > 0$ and Lipschitz with constant $c_{\text{lip}} > 0$. For example, in visual-servoing, (SOC1) can be formulated as a *goal tracking* problem (Ebert et al., 2018), where we control the robot to reach the goal observation $x_{\text{goal}}$, and the objective is to compute a sequence of optimal open-loop actions $U$ that minimizes the cumulative tracking error $\mathbb{E}[\sum_t \|x_t - x_{\text{goal}}\|^2 \mid P, x_0]$.

Since the observations $x$ are high dimensional and the dynamics in the observation space $P(\cdot|x_t, u_t)$ is unknown, solving (SOC1) is often intractable. To address this issue, a class of algorithms has been recently developed that is based on learning a low-dimensional latent (embedding) space $\mathcal{Z} \subseteq \mathbb{R}^{n_z}$ ($n_z \ll n_x$) and latent state dynamics, and performing optimal control there. This class that we refer to as *learning controllable embedding* (LCE) throughout the paper, include recently developed algorithms, such as E2C (Watter et al., 2015), RCE (Banijamali et al., 2018), and SOLAR (Zhang et al., 2019). The main idea behind the LCE approach is to learn a triplet, (i) an encoder $E : \mathcal{X} \to \mathbb{P}(\mathcal{Z})$; (ii) a dynamics in the latent space $F : \mathcal{Z} \times \mathcal{U} \to \mathbb{P}(\mathcal{Z})$; and (iii) a decoder $D : \mathcal{Z} \to \mathbb{P}(\mathcal{X})$. These in turn can be thought of as defining a (stochastic) mapping $\widehat{P} : \mathcal{X} \times \mathcal{U} \to \mathbb{P}(\mathcal{X})$ of the form $\widehat{P} = D \circ F \circ E$. We then wish to solve the SOC in latent space $\mathcal{Z}$:

$$\min_{U, \widehat{P}} \quad \mathbb{E}\Big[L(U, F, \overline{c}, z_0) \mid E, x_0\Big] + \lambda_2 \sqrt{R_2(\widehat{P})}, \tag{SOC2}$$

such that the solution of (SOC2), $U_2^*$, has similar performance to that of (SOC1), $U_1^*$, i.e., $L(U_1^*, P, c, x_0) \approx L(U_2^*, P, c, x_0)$. In (SOC2), $z_0$ is the initial latent state sampled from the encoder $E(\cdot|x_0)$; $\overline{c} : \mathcal{Z} \times \mathcal{U} \to \mathbb{R}_{\geq 0}$ is the latent cost function defined as $\overline{c}_t(z_t, u_t) = \int c_t(x_t, u_t) dD(x_t|z_t)$; $R_2(\widehat{P})$ is a regularizer over the mapping $\widehat{P}$; and $\lambda_2$ is the corresponding

---

[1]A method to ensure this Markovian assumption is by buffering observations (Mnih et al., 2013) for a number of time steps.

[2]See Appendix B.3 for the extension to the closed-loop MDP problem.

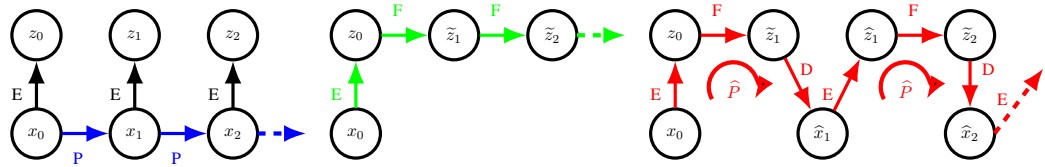

Figure 1: Evolution of the states (a)(blue) in equation SOC1 under dynamics $P$, (b)(green) in equation SOC2 under dynamics $F$, and (c)(red) in equation SOC3 under dynamics $\widehat{P}$.

regularization parameter. We will define $R_2$ and $\lambda_2$ more precisely in Section 3. Note that the expectation in (SOC2) is over the randomness generated by the (stochastic) encoder $E$.

## 3  PCC MODEL: A CONTROL PERSPECTIVE

As described in Section 2, we are primarily interested in solving (SOC1), whose states evolve under dynamics $P$, as shown at the bottom row of Figure 1(a) in (blue). However, because of the difficulties in solving (SOC1), mainly due to the high dimension of observations $x$, LCE proposes to learn a mapping $\widehat{P}$ by solving (SOC2) that consists of a loss function, whose states evolve under dynamics $F$ (after an initial transition by encoder $E$), as depicted in Figure 1(b), and a regularization term. The role of the regularizer $R_2$ is to account for the performance gap between (SOC1) and the loss function of (SOC2), due to the discrepancy between their evolution paths, shown in Figures 1(a)(blue) and 1(b)(green). The goal of LCE is to learn $\widehat{P}$ of the particular form $\widehat{P} = D \circ F \circ E$, described in Section 2, such that the solution of (SOC2) has similar performance to that of (SOC1). In this section, we propose a principled way to select the regularizer $R_2$ to achieve this goal. Since the exact form of (SOC2) has a direct effect on learning $\widehat{P}$, designing this regularization term, in turn, provides us with a recipe (loss function) to learn the latent (embedded) space $\mathcal{Z}$. In the following subsections, we show that this loss function consists of three terms that correspond to *prediction*, *consistency*, and *curvature*, the three ingredients of our PCC model.

Note that these two SOCs evolve in two different spaces, one in the observation space $\mathcal{X}$ under dynamics $P$, and the other one in the latent space $\mathcal{Z}$ (after an initial transition from $\mathcal{X}$ to $\mathcal{Z}$) under dynamics $F$. Unlike $P$ and $F$ that only operate in a single space, $\mathcal{X}$ and $\mathcal{Z}$, respectively, $\widehat{P}$ can govern the evolution of the system in both $\mathcal{X}$ and $\mathcal{Z}$ (see Figure 1(c)). Therefore, any recipe to learn $\widehat{P}$, and as a result the latent space $\mathcal{Z}$, should have at least two terms, to guarantee that the evolution paths resulted from $\widehat{P}$ in $\mathcal{X}$ and $\mathcal{Z}$ are consistent with those generated by $P$ and $F$. We derive these two terms, that are the *prediction* and *consistency* terms in the loss function used by our PCC model, in Sections 3.1 and 3.2, respectively. While these two terms are the result of learning $\widehat{P}$ in general SOC problems, in Section 3.3, we concentrate on the particular class of LLC algorithms (e.g., iLQR (Li & Todorov, 2004)) to solve SOC, and add the third term, *curvature*, to our recipe for learning $\widehat{P}$.

### 3.1  PREDICTION OF THE NEXT OBSERVATION

Figures 1(a)(blue) and 1(c)(red) show the transition in the observation space under $P$ and $\widehat{P}$, where $x_t$ is the current observation, and $x_{t+1}$ and $\hat{x}_{t+1}$ are the next observations under these two dynamics, respectively. Instead of learning a $\widehat{P}$ with minimum mismatch with $P$ in terms of some distribution norm, we propose to learn $\widehat{P}$ by solving the following SOC:

$$\min_{U,\widehat{P}} \quad L(U, \widehat{P}, c, x_0) + \lambda_3 \sqrt{R_3(\widehat{P})}, \tag{SOC3}$$

whose loss function is the same as the one in (SOC1), with the true dynamics replaced by $\widehat{P}$. In Lemma 1 (see Appendix A.1, for proof), we show how to set the regularization term $R_3$ in (SOC3), such that the control sequence resulted from solving (SOC3), $U_3^*$, has similar performance to the solution of (SOC1), $U_1^*$, i.e., $L(U_1^*, P, c, x_0) \approx L(U_3^*, P, c, x_0)$.

**Lemma 1.** *Let $U_1^*$ be a solution to (SOC1) and $(U_3^*, \widehat{P}_3^*)$ be a solution to (SOC3) with*

$$R_3(\widehat{P}) = \mathbb{E}_{x,u}\big[D_{KL}\big(P(\cdot|x,u)||\widehat{P}(\cdot|x,u)\big)\big] \quad and \quad \lambda_3 = \sqrt{2\overline{U}} \cdot T^2 c_{max}. \tag{1}$$

*Then, we have $L(U_1^*, P, c, x_0) \geq L(U_3^*, P, c, x_0) - 2\lambda_3 \sqrt{R_3(\widehat{P}_3^*)}$.*

In Eq. 1, the expectation is over the state-action stationary distribution of the policy used to generate the training samples (uniformly random policy in this work), and $\overline{U}$ is the *Lebesgue measure* of $\mathcal{U}$.[3]

---

[3]In the case when sampling policy is non-uniform and has no measure-zero set, $1/\overline{U}$ is its minimum measure.

## 3.2 Consistency in Prediction of the Next Latent State

In Section 3.1, we provided a recipe for learning $\widehat{P}$ (in form of $D \circ F \circ E$) by introducing an intermediate (SOC3) that evolves in the observation space $\mathcal{X}$ according to dynamics $\widehat{P}$. In this section we first connect (SOC2) that operates in $\mathcal{Z}$ with (SOC3) that operates in $\mathcal{X}$. For simplicity and without loss generality, assume the initial cost $c_0(x, u)$ is zero.[4] Lemma 2 (see Appendix A.2, for proof) suggests how we shall set the regularizer in (SOC2), such that its solution performs similarly to that of (SOC3), under their corresponding dynamics models.

**Lemma 2.** *Let $(U_3^*, \widehat{P}_3^*)$ be a solution to (SOC3) and $(U_2^*, \widehat{P}_2^*)$ be a solution to (SOC2) with*

$$R_2'(\widehat{P}) = \mathbb{E}_{x,u}\Big[D_{KL}\big((E \circ \widehat{P})(\cdot|x,u)||(F \circ E)(\cdot|x,u)\big)\Big] \qquad and \qquad \lambda_2 = \sqrt{2\overline{U}} \cdot T^2 c_{max}. \quad (2)$$

*Then, we have $L(U_3^*, \widehat{P}_3^*, c, x_0) \geq L(U_2^*, \widehat{P}_2^*, c, x_0) - 2\lambda_2\sqrt{R_2'(\widehat{P}_2^*)}$ .*

Similar to Lemma 1, in Eq. 2, the expectation is over the state-action stationary distribution of the policy used to generate the training samples. Moreover, $(E \circ \widehat{P})(z'|x,u) = \int_{x'} E(z'|x')d\widehat{P}(x'|x,u)$ and $(F \circ E)(z'|x,u) = \int_z F(z'|z,u)dE(z|x)$ are the probability over the next latent state $z'$, given the current observation $x$ and action $u$, in (SOC2) and (SOC3) (see the paths $x_t \to z_t \to \tilde{z}_{t+1}$ and $x_t \to z_t \to \tilde{z}_{t+1} \to \hat{x}_{t+1} \to \hat{z}_{t+1}$ in Figures 1(b)(green) and 1(c)(red)). Therefore $R_2'(\widehat{P})$ can be interpreted as the measure of discrepancy between these models, which we term as *consistency* loss.

Although Lemma 2 provides a recipe to learn $\widehat{P}$ by solving (SOC2) with the regularizer (2), unfortunately this regularizer cannot be computed from the data – that is of the form $(x_t, u_t, x_{t+1})$ – because the first term in the $D_{KL}$ requires marginalizing over current and next latent states ($z_t$ and $\tilde{z}_{t+1}$ in Figure 1(c)). To address this issue, we propose to use the (computable) regularizer

$$R_2''(\widehat{P}) = \mathbb{E}_{x,u,x'}\Big[D_{KL}\big(E(\cdot|x')||(F \circ E)(\cdot|x,u)\big)\Big], \quad (3)$$

in which the expectation is over $(x, u, x')$ sampled from the training data. Corollary 1 (see Appendix A.3, for proof) bounds the performance loss resulted from using $R_2''(\widehat{P})$ instead of $R_2'(\widehat{P})$, and shows that it could be still a reasonable choice.

**Corollary 1.** *Let $(U_3^*, \widehat{P}_3^*)$ be a solution to (SOC3) and $(U_2^*, \widehat{P}_2^*)$ be a solution to (SOC2) with $R_2''(\widehat{P})$ and and $\lambda_2$ defined by (3) and (2). Then, we have $L(U_3^*, \widehat{P}_3^*, c, x_0) \geq L(U_2^*, \widehat{P}_2^*, c, x_0) - 2\lambda_2\sqrt{2R_2''(\widehat{P}_2^*) + 2R_3(\widehat{P}_2^*)}$ .*

Lemma 1 suggests a regularizer $R_3$ to connect the solutions of (SOC1) and (SOC3). Similarly, Corollary 1 shows that regularizer $R_2''$ in (3) establishes a connection between the solutions of (SOC3) and (SOC2). Putting these results together, we achieve our goal in Lemma 3 (see Appendix A.4, for proof) to design a regularizer for (SOC2), such that its solution performs similarly to that of (SOC1).

**Lemma 3.** *Let $U_1^*$ be a solution to (SOC1) and $(U_2^*, \widehat{P}_2^*)$ be a solution to (SOC2) with*

$$R_2(\widehat{P}) = 3R_3(\widehat{P}) + 2R_2''(\widehat{P}) \qquad and \qquad \lambda_2 = 2\sqrt{\overline{U}} \cdot T^2 c_{max}, \quad (4)$$

*where $R_3(\widehat{P})$ and $R_2''(\widehat{P})$ are defined by (1) and (3). Then, we have*

$$L(U_1^*, P, c, x_0) \geq L(U_2^*, P, c, x_0) - 2\lambda_2\sqrt{R_2(\widehat{P}_2^*)} .$$

## 3.3 Locally-Linear Control in the Latent Space and Curvature Regularization

In Sections 3.1 and 3.2, we derived a loss function to learn the latent space $\mathcal{Z}$. This loss function, that was motivated by the general SOC perspective, consists of two terms to enforce the latent space to not only predict the next observations accurately, but to be suitable for control. In this section, we focus on the class of locally-linear control (LLC) algorithms (e.g., iLQR), for solving (SOC2), and show how this choice adds a third term, that corresponds to *curvature*, to the regularizer of (SOC2), and as a result, to the loss function of our PCC model.

The main idea in LLC algorithms is to iteratively compute an action sequence to improve the current trajectory, by linearizing the dynamics around this trajectory, and use this action sequence to generate

---

[4]With non-zero initial cost, similar results can be derived by having an additional consistency term on $x_0$.

the next trajectory (see Appendix B for more details about LLC and iLQR). This procedure implicitly assumes that the dynamics is approximately locally linear. To ensure this in (SOC2), we further restrict the dynamics $\widehat{P}$ and assume that it is not only of the form $\widehat{P} = D \circ F \circ E$, but $F$, the latent space dynamics, has low curvature. One way to ensure this in (SOC2) is to directly impose a penalty over the curvature of the latent space transition function $f_{\mathcal{Z}}(z, u)$. Assume $F(z, u) = f_{\mathcal{Z}}(z, u) + w$, where $w$ is a Gaussian noise. Consider the following SOC problem:

$$\min_{U, \widehat{P}} \quad \mathbb{E}\left[L(U, F, \overline{c}, z_0) \mid E, x_0\right] + \lambda_{\text{LLC}}\sqrt{R_2(\widehat{P}) + R_{\text{LLC}}(\widehat{P})}, \tag{SOC-LLC}$$

where $R_2$ is defined by (4); $U$ is optimized by a LLC algorithm, such as iLQR; $R_{\text{LLC}}(\widehat{P})$ is given by,

$$R_{\text{LLC}}(\widehat{P}) = \mathbb{E}_{x,u}\left[\mathbb{E}_{\epsilon}\left[f_{\mathcal{Z}}(z + \epsilon_z, u + \epsilon_u) - f_{\mathcal{Z}}(z, u) - (\nabla_z f_{\mathcal{Z}}(z, u) \cdot \epsilon_z + \nabla_u f_{\mathcal{Z}}(z, u) \cdot \epsilon_u)\|_2^2\right] \mid E\right], \tag{5}$$

where $\epsilon = (\epsilon_z, \epsilon_u)^\top \sim \mathcal{N}(0, \delta^2 I)$, $\delta > 0$ is a tunable parameter that characterizes the "diameter" of latent state-action space in which the latent dynamics model has low curvature. $\lambda_{\text{LLC}} = 2\sqrt{2}T^2 c_{\max}\sqrt{\overline{U}}\max\left(c_{\text{lip}}(1 + \sqrt{2\log(2T/\eta)})\sqrt{\overline{X}}/2, 1\right)$, where $1/\overline{X}$ is the minimum non-zero measure of the sample distribution w.r.t. $\mathcal{X}$, and $1 - \eta \in [0, 1)$ is a probability threshold. Lemma 4 (see Appendix A.5, for proof and discussions on how $\delta$ affects LLC performance) shows that a solution of (SOC-LLC) has similar performance to a solution of (SOC1), and thus, (SOC-LLC) is a reasonable optimization problem to learn $\widehat{P}$, and also the latent space $\mathcal{Z}$.

**Lemma 4.** *Let $(U^*_{LLC}, \widehat{P}^*_{LLC})$ be a LLC solution to (SOC-LLC) and $U^*_1$ be a solution to (SOC1). Suppose the nominal latent state-action trajectory $\{(\mathbf{z}_t, \mathbf{u}_t)\}_{t=0}^{T-1}$ satisfies the condition: $(\mathbf{z}_t, \mathbf{u}_t) \sim \mathcal{N}((z^*_{2,t}, u^*_{2,t}), \delta^2 I)$, where $\{(z^*_{2,t}, u^*_{2,t})\}_{t=0}^{T-1}$ is the optimal trajectory of (SOC2). Then with probability $1 - \eta$, we have $L(U^*_1, P, c, x_0) \geq L(U^*_{LLC}, P, c, x_0) - 2\lambda_{LLC}\sqrt{R_2(\widehat{P}^*_{LLC}) + R_{LLC}(\widehat{P}^*_{LLC})}$.*

In practice, instead of solving (SOC-LLC) jointly for $U$ and $\widehat{P}$, we treat (SOC-LLC) as a bi-level optimization problem, first, solve the inner optimization problem for $\widehat{P}$, i.e.,

$$\widehat{P}^* \in \arg\min_{\widehat{P}} \lambda_{\text{p}}R'_3(\widehat{P}) + \lambda_{\text{c}}R''_2(\widehat{P}) + \lambda_{\text{cur}}R_{\text{LLC}}(\widehat{P}), \tag{PCC-LOSS}$$

where $R'_3(\widehat{P}) = -\mathbb{E}_{x,u,x'}[\log \widehat{P}(x'|x, u)]$ is the *negative log-likelihood*,[5] and then, solve the outer optimization problem, $\min_U L(U, \widehat{F}^*, \overline{c}, z_0)$, where $\widehat{P}^* = \widehat{D}^* \circ \widehat{F}^* \circ \widehat{E}^*$, to obtain the optimal control sequence $U^*$. Solving (SOC-LLC) this way is an approximation, in general, but is justified, when the regularization parameter $\lambda_{\text{LLC}}$ is large. Note that we leave the regularization parameters $(\lambda_{\text{p}}, \lambda_{\text{c}}, \lambda_{\text{cur}})$ as hyper-parameters of our algorithm, and do not use those derived in the lemmas of this section. Since the loss for learning $\widehat{P}^*$ in (PCC-LOSS) enforces (i) prediction accuracy, (ii) consistency in latent state prediction, and (iii) low curvature over $f_{\mathcal{Z}}$, through the regularizers $R'_3$, $R''_2$, and $R_{\text{LLC}}$, respectively, we refer to it as the *prediction-consistency-curvature* (PCC) loss.

## 4 INSTANTIATING THE PCC MODEL IN PRACTICE

The PCC-Model objective in (PCC-LOSS) introduces the optimization problem $\min_{\widehat{P}} \lambda_{\text{p}}R'_3(\widehat{P}) + \lambda_{\text{c}}R''_2(\widehat{P}) + \lambda_{\text{cur}}R_{\text{LLC}}(\widehat{P})$. To instantiate this model in practice, we describe $\widehat{P} = D \circ F \circ E$ as a latent variable model that factorizes as $\widehat{P}(x_{t+1}, z_t, \hat{z}_{t+1} \mid x_t, u_t) = \widehat{P}(z_t \mid x_t)\widehat{P}(\hat{z}_{t+1} \mid z_t, u_t)\widehat{P}(x_{t+1} \mid \hat{z}_{t+1})$. In this section, we propose a variational approximation to the intractable negative log-likelihood $R'_3$ and batch-consistency $R''_2$ losses, and an efficient approximation of the curvature loss $R_{\text{LLC}}$.

### 4.1 VARIATIONAL PCC

The negative log-likelihood [6] $R'_3$ admits a variational bound via Jensen's Inequality,

$$R'_3(\widehat{P}) = -\log \widehat{P}(x_{t+1} \mid x_t, u_t) = -\log \mathbb{E}_{Q(z_t, \hat{z}_{t+1}|x_t, u_t, x_{t+1})}\left[\frac{\widehat{P}(x_{t+1}, z_t, \hat{z}_{t+1} \mid x_t, u_t)}{Q(z_t, \hat{z}_{t+1} \mid x_t, u_t, x_{t+1})}\right]$$

$$\leq -\mathbb{E}_{Q(z_t, \hat{z}_{t+1}|x_t, u_t, x_{t+1})}\left[\log \frac{\widehat{P}(x_{t+1}, z_t, \hat{z}_{t+1} \mid x_t, u_t)}{Q(z_t, \hat{z}_{t+1} \mid x_t, u_t, x_{t+1})}\right] = R'_{3,\text{NLE-Bound}}(\widehat{P}, Q), \tag{6}$$

---

[5]Since $R_3(\widehat{P})$ is the sum of $R'_3(\widehat{P})$ and the entropy of $P$, we replaced it with $R'_3(\widehat{P})$ in (PCC-LOSS).

[6]For notation convenience, we drop the expectation over the empirical data that appears in various loss terms.

which holds for any choice of recognition model $Q$. For simplicity, we assume the recognition model employs bottom-up inference and thus factorizes as $Q(z_t, \hat{z}_{t+1}|x_t, x_{t+1}, u_t) = Q(\hat{z}_{t+1}|x_{t+1})Q(z_t|\hat{z}_{t+1}, x_t, u_t)$. The main idea behind choosing a backward-facing model is to allow the model to learn to account for noise in the underlying dynamics. We estimate the expectations in (6) via Monte Carlo simulation. To reduce the variance of the estimator, we decompose $R'_{3,\text{NLE-Bound}}$ further into

$$- \mathbb{E}_{Q(\hat{z}_{t+1}|x_{t+1})}\left[\log \widehat{P}(x_{t+1}|\hat{z}_{t+1})\right] + \mathbb{E}_{Q(\hat{z}_{t+1}|x_{t+1})}\left[D_{\text{KL}}\left(Q(z_t \mid \hat{z}_{t+1}, x_t, u_t)\|\widehat{P}(z_t \mid x_t)\right)\right]$$

$$- H\left(Q(\hat{z}_{t+1} \mid x_{t+1})\right) - \mathbb{E}_{\substack{Q(\hat{z}_{t+1}|x_{t+1})\\Q(z_t|\hat{z}_{t+1}, x_t, u_t)}}\left[\log \widehat{P}(\hat{z}_{t+1} \mid z_t, u_t)\right],$$

and note that the Entropy $H(\cdot)$ and Kullback-Leibler $D_{\text{KL}}(\cdot\|\cdot)$ terms are analytically tractable when $Q$ is restricted to a suitably chosen variational family (i.e. in our experiments, $Q(\hat{z}_{t+1} \mid x_{t+1})$ and $Q(z_t \mid \hat{z}_{t+1}, x_t, u_t)$ are factorized Gaussians). The derivation is provided in Appendix C.1.

Interestingly, the consistency loss $R''_2$ admits a similar treatment. We note that the consistency loss seeks to match the distribution of $\hat{z}_{t+1} \mid x_t, u_t$ with $z_{t+1} \mid x_{t+1}$, which we represent below as

$$R''_2(\widehat{P}) = D_{\text{KL}}\left(\widehat{P}(z_{t+1} \mid x_{t+1})\|\widehat{P}(\hat{z}_{t+1} \mid x_t, u_t)\right) = -H(\widehat{P}(z_{t+1} \mid x_{t+1})) - \mathbb{E}_{\substack{\widehat{P}(z_{t+1}|x_{t+1})\\\hat{z}_{t+1}=z_{t+1}}}\left[\log \widehat{P}(\hat{z}_{t+1} \mid x_t, u_t)\right].$$

Here, $\widehat{P}(\hat{z}_{t+1} \mid x_t, u_t)$ is intractable due to the marginalization of $z_t$. We employ the same procedure as in (6) to construct a tractable variational bound

$$R''_2(\widehat{P}) \leq -H(\widehat{P}(z_{t+1} \mid x_{t+1})) - \mathbb{E}_{\substack{\widehat{P}(z_{t+1}|x_{t+1})\\\hat{z}_{t+1}=z_{t+1}}}\mathbb{E}_{Q(z_t|\hat{z}_{t+1}, x_t, u_t)}\left[\log \frac{\widehat{P}(z_t, \hat{z}_{t+1} \mid x_t, u_t)}{Q(z_t \mid \hat{z}_{t+1}, x_t, u_t)}\right].$$

We now make the further simplifying assumption that $Q(\hat{z}_{t+1} \mid x_{t+1}) = \widehat{P}(\hat{z}_{t+1} \mid x_{t+1})$. This allows us to rewrite the expression as

$$R''_2(\widehat{P}) \leq -H(Q(\hat{z}_{t+1} \mid x_{t+1})) - \mathbb{E}_{\substack{Q(\hat{z}_{t+1}|x_{t+1})\\Q(z_t|\hat{z}_{t+1}, x_t, u_t)}}\left[\log \widehat{P}(\hat{z}_{t+1} \mid z_t, u_t)\right]$$

$$+ \mathbb{E}_{Q(\hat{z}_{t+1}|x_{t+1})}\left[D_{\text{KL}}(Q(z_t \mid \hat{z}_{t+1}, x_t, u_t)\|\widehat{P}(z_t \mid x_t))\right] = R''_{2,\text{Bound}}(\widehat{P}, Q), \tag{7}$$

which is a subset of the terms in (6). See Appendix C.2 for a detailed derivation.

## 4.2 Curvature Regularization and Amortized Gradient

In practice we use a variant of the curvature loss where Taylor expansions and gradients are evaluated at $\bar{z} = z + \epsilon_z$ and $\bar{u} = u + \epsilon_u$,

$$R_{\text{LLC}}(\widehat{P}) = \mathbb{E}_{\epsilon \sim \mathcal{N}(0, \delta I)}[\|f_{\mathcal{Z}}(\bar{z}, \bar{u}) - (\nabla_z f_{\mathcal{Z}}(\bar{z}, \bar{u})\epsilon_z + \nabla_u f_{\mathcal{Z}}(\bar{z}, \bar{u})\epsilon_u) - f_{\mathcal{Z}}(z, u)\|_2^2]. \tag{8}$$

When $n_z$ is large, evaluation and differentiating through the Jacobians can be slow. To circumvent this issue, the Jacobians evaluation can be amortized by treating the Jacobians as the coefficients of the best linear approximation at the evaluation point. This leads to a new *amortized* curvature loss

$$R_{\text{LLC-Amor}}(\widehat{P}, A, B) = \mathbb{E}_{\epsilon \sim \mathcal{N}(0, \delta I)}[\|f_{\mathcal{Z}}(\bar{z}, \bar{u}) - (A(\bar{z}, \bar{u})\epsilon_z + B(\bar{z}, \bar{u})\epsilon_u - f_{\mathcal{Z}}(z, u))\|_2^2]. \tag{9}$$

where $A$ and $B$ are function approximators to be optimized. Intuitively, the amortized curvature loss seeks—for any given $(z, u)$—to find the best choice of linear approximation induced by $A(z, u)$ and $B(z, u)$ such that the behavior of $F_\mu$ in the neighborhood of $(z, u)$ is approximately linear.

## 5 Relation to Previous Embed-to-Control Approaches

In this section, we highlight the key differences between PCC and the closest previous works, namely E2C and RCE. A key distinguishing factor is PCC's use of a nonlinear latent dynamics model paired with an explicit curvature loss. In comparison, E2C and RCE both employed "locally-linear dynamics" of the form $z' = A(\bar{z}, \bar{u})z + B(\bar{z}, \bar{u})u + c(\bar{z}, \bar{u})$ where $\bar{z}$ and $\bar{u}$ are auxiliary random variables meant to be perturbations of $z$ and $u$. When contrasted with (9), it is clear that neither $A$ and $B$ in the E2C/RCE formulation can be treated as the Jacobians of the dynamics, and hence the curvature of the dynamics is not being controlled explicitly. Furthermore, since the locally-linear dynamics are wrapped inside the maximum-likelihood estimation, both E2C and RCE conflate the two key elements prediction and curvature together. This makes controlling the stability of training much more difficult. Not only does PCC explicitly separate these two components, we are also the first to explicitly demonstrate theoretically and empirically that the curvature loss is important for iLQR.

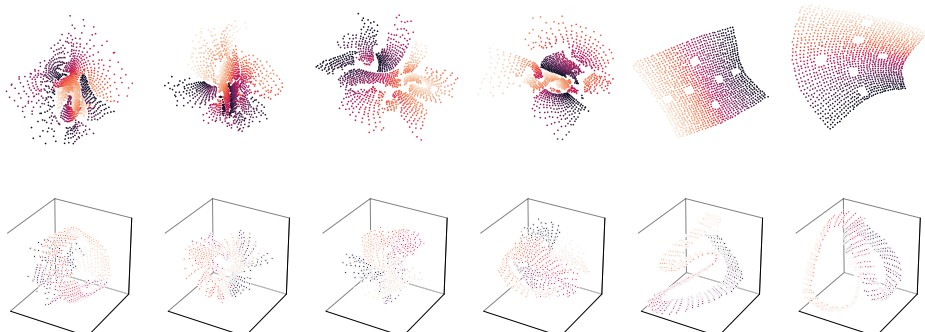

Figure 2: Top: Planar latent representations; Bottom: Inverted Pendulum latent representations (randomly selected): left two: RCE, middle two: E2C, right two: PCC.

Furthermore, RCE does not incorporate PCC's consistency loss. Note that PCC, RCE, and E2C are all Markovian encoder-transition-decoder frameworks. Under such a framework, the sole reliance on minimizing the prediction loss will result in a discrepancy between how the model is trained (maximizing the likelihood induced by encoding-transitioning-decoding) versus how it is used at test-time for control (continual transitioning in the latent space without ever decoding). By explicitly minimizing the consistency loss, PCC reduces the discrepancy between how the model is trained versus how it is used at test-time for planning. Interestingly, E2C does include a regularization term that is akin to PCC's consistency loss. However, as noted by the authors of RCE, E2C's maximization of pair-marginal log-likelihoods of $(x_t, x_{t+1})$ as opposed to the conditional likelihood of $x_{t+1}$ given $x_t$ means that E2C does not properly minimize the prediction loss prescribed by the PCC framework.

## 6 EXPERIMENTS

In this section, we compare the performance of PCC with two model-based control algorithm baselines: RCE[7] (Banijamali et al., 2018) and E2C (Watter et al., 2015), as well as running a thorough ablation study on various components of PCC. The experiments are based on the following continuous control benchmark domains (see Appendix D for more descriptions): (i) Planar System, (ii) Inverted Pendulum, (iii) Cartpole, (iv) 3-link manipulator, and (v) TORCS simulator[8] (Wymann et al., 2000).

To generate our training and test sets, each consists of triples $(x_t, u_t, x_{t+1})$, we: (1) sample an underlying state $s_t$ and generate its corresponding observation $x_t$, (2) sample an action $u_t$, and (3) obtain the next state $s_{t+1}$ according to the state transition dynamics, add it a zero-mean Gaussian noise with variance $\sigma^2 I_{n_s}$, and generate corresponding observation $x_{t+1}$. To ensure that the observation-action data is uniformly distributed (see Section 3), we sample the state-action pair $(s_t, u_t)$ uniformly from the state-action space. To understand the robustness of each model, we consider both deterministic ($\sigma = 0$) and stochastic scenarios. In the stochastic case, we add noise to the system with different values of $\sigma$ and evaluate the models' performance under various degree of noise.

Each task has underlying start and goal states that are unobservable to the algorithms, instead, the algorithms have access to the corresponding start and goal observations. We apply control using the iLQR algorithm (see Appendix B), with the same cost function that was used by RCE and E2C, namely, $\bar{c}(z_t, u_t) = (z_t - z_{\text{goal}})^\top Q(z_t - z_{\text{goal}}) + u_t^\top R u_t$, and $\bar{c}(z_T) = (z_T - z_{\text{goal}})^\top Q(z_T - z_{\text{goal}})$, where $z_{\text{goal}}$ is obtained by encoding the goal observation, and $Q = \kappa \cdot I_{n_z}$, $R = I_{n_u}$[9]. Details of our implementations are specified in Appendix D.3. We report performance in the underlying system, specifically the percentage of time spent in the goal region[10].

**A Reproducible Experimental Pipeline** In order to measure performance reproducibility, we perform the following 2-step pipeline. For each control task and algorithm, we (1) train 10 models

---

[7]For the RCE implementation, we directly optimize the ELBO loss in Equation (16) of the paper. We also tried the approach reported in the paper on increasing the weights of the two middle terms and then annealing them to 1. However, in practice this method is sensitive to annealing schedule and has convergence issues.

[8]See a control demo on the TORCS simulator at https://youtu.be/GBrgALRZ2fw

[9]According to the definition of latent cost $\bar{c}(z, u) = D \circ c(z, u)$, its quadratic approximation is given by

$$\bar{c}(z, u) \approx \begin{bmatrix} z - z_{\text{goal}} \\ u \end{bmatrix}^\top \begin{bmatrix} \nabla_z \\ \nabla_u \end{bmatrix} D \circ c|_{z=z_{\text{goal}}, u=0} + \frac{1}{2} \begin{bmatrix} z - z_{\text{goal}} \\ u \end{bmatrix}^\top \begin{bmatrix} \nabla_{zz}^2 & \nabla_{zu}^2 \\ \nabla_{uz}^2 & \nabla_{uu}^2 \end{bmatrix} D \circ c|_{z=z_{\text{goal}}, u=0} \begin{bmatrix} z - z_{\text{goal}} \\ u \end{bmatrix}.$$

Yet for simplicity, we choose the same latent cost as in RCE and E2C with fixed, tunable matrices $Q$ and $R$.

[10]Another possible metric is the average distance to goal, which has a similar behavior.

Table 1: Percentage of steps in goal state. Averaged over all models (left), and the best model (right).

| Domain | RCE (all) | E2C (all) | PCC (all) | RCE (top 1) | E2C (top 1) | PCC (top 1) |
|--------|-----------|-----------|-----------|-------------|-------------|-------------|
| Planar | $2.1 \pm 0.8$ | $5.5 \pm 1.7$ | $\mathbf{35.7 \pm 3.4}$ | $9.2 \pm 1.4$ | $36.5 \pm 3.6$ | $\mathbf{72.1 \pm 0.4}$ |
| Pendulum | $24.7 \pm 3.1$ | $46.8 \pm 4.1$ | $\mathbf{58.7 \pm 3.7}$ | $68.8 \pm 2.2$ | $89.7 \pm 0.5$ | $\mathbf{90.3 \pm 0.4}$ |
| Cartpole | $\mathbf{59.5 \pm 4.1}$ | $7.3 \pm 1.5$ | $54.3 \pm 3.9$ | $\mathbf{99.45 \pm 0.1}$ | $40.2 \pm 3.2$ | $93.9 \pm 1.7$ |
| 3-link | $1.1 \pm 0.4$ | $4.7 \pm 1.1$ | $\mathbf{18.8 \pm 2.1}$ | $10.6 \pm 0.8$ | $20.9 \pm 0.8$ | $\mathbf{47.2 \pm 1.7}$ |
| TORCS | $27.4 \pm 1.8$ | $28.2 \pm 1.9$ | $\mathbf{60.7 \pm 1.1}$ | $39.9 \pm 2.2$ | $54.1 \pm 2.3$ | $\mathbf{68.6 \pm 0.4}$ |

Table 2: Ablation analysis. Percentage of steps spent in goal state. From left to right: PCC including all loss terms, excluding consistency loss, excluding curvature loss, amortizing the curvature loss.

| Domain | PCC | PCC no Con | PCC no Cur | PCC Amor |
|--------|-----|------------|------------|----------|
| Planar | $35.7 \pm 3.4$ | $0.0 \pm 0.0$ | $29.6 \pm 3.5$ | $\mathbf{41.7 \pm 3.7}$ |
| Pendulum | $\mathbf{58.7 \pm 3.7}$ | $52.3 \pm 3.5$ | $50.3 \pm 3.3$ | $54.2 \pm 3.1$ |
| Cartpole | $\mathbf{54.3 \pm 3.9}$ | $5.1 \pm 0.4$ | $17.4 \pm 1.6$ | $14.3 \pm 1.2$ |
| 3-link | $\mathbf{18.8 \pm 2.1}$ | $9.1 \pm 1.5$ | $13.1 \pm 1.9$ | $11.5 \pm 1.8$ |

independently, and (2) solve 10 control tasks per model (we **do not** cherry-pick, but instead perform a total of $10 \times 10 = 100$ control tasks). We report statistics averaged over **all** the tasks (in addition, we report the best performing model averaged over its 10 tasks). By adopting a principled and statistically reliable evaluation pipeline, we also address a pitfall of the compared baselines where the best model needs to be cherry picked, and training variance was not reported.

**Results**   Table 1 shows how PCC outperforms the baseline algorithms in the noiseless dynamics case by comparing means and standard deviations of the means on the different control tasks (for the case of added noise to the dynamics, which exhibits similar behavior, refer to Appendix E.1). It is important to note that for each algorithm, the performance metric averaged over all models is drastically different than that of the best model, which justifies our rationale behind using the reproducible evaluation pipeline and avoid cherry-picking when reporting. Figure 2 depicts 2 instances (randomly chosen from the 10 trained models) of the learned latent space representations on the noiseless dynamics of Planar and Inverted Pendulum tasks for PCC, RCE, and E2C models (additional representations can be found in Appendix E.2). Representations were generated by encoding observations corresponding to a uniform grid over the state space. Generally, PCC has a more interpretable representation of both Planar and Inverted Pendulum Systems than other baselines for both the noiseless dynamics case and the noisy case. Finally, in terms of computation, PCC demonstrates faster training with 64% improvement over RCE, and 2% improvement over E2C.[11]

**Ablation Analysis**   On top of comparing the performance of PCC to the baselines, in order to understand the importance of each component in (PCC-LOSS), we also perform an ablation analysis on the consistency loss (with/without consistency loss) and the curvature loss (with/without curvature loss, and with/without amortization of the Jacobian terms). Table 2 shows the ablation analysis of PCC on the aforementioned tasks. From the numerical results, one can clearly see that when consistency loss is omitted, the control performance degrades. This corroborates with the theoretical results in Section 3.2, which indicates the relationship of the consistency loss and the estimation error between the next-latent dynamics prediction and the next-latent encoding. This further implies that as the consistency term vanishes, the gap between control objective function and the model training loss is widened, due to the accumulation of state estimation error. The control performance also decreases when one removes the curvature loss. This is mainly attributed to the error between the iLQR control algorithm and (SOC2). Although the latent state dynamics model is parameterized with neural networks, which are smooth, without enforcing the curvature loss term the norm of the Hessian (curvature) might still be high. This also confirms with the analysis in Section 3.3 about sub-optimality performance and curvature of latent dynamics. Finally, we observe that the performance of models trained without amortized curvature loss are slightly better than with their amortized counterpart, however, since the amortized curvature loss does not require computing gradient of the latent dynamics (which means that in stochastic optimization one does not need to estimate its Hessian), we observe relative speed-ups in model training with the amortized version (speed-up of 6%, 9%, and 15% for Planar System, Inverted Pendulum, and Cartpole, respectively).

# 7 CONCLUSION

In this paper, we argue from first principles that learning a latent representation for control should be guided by good prediction in the observation space and consistency between latent transition and

---

[11]Comparison jobs were deployed on the Planar system using Nvidia TITAN Xp GPU.

the embedded observations. Furthermore, if variants of iterative LQR are used as the controller, the low-curvature dynamics is desirable. All three elements of our PCC models are critical to the stability of model training and the performance of the in-latent-space controller. We hypothesize that each particular choice of controller will exert different requirement for the learned dynamics. A future direction is to identify and investigate the additional bias for learning an effective embedding and latent dynamics for other type of model-based control and planning methods.

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

# A    TECHNICAL PROOFS OF SECTION 3

## A.1    PROOF OF LEMMA 1

Following analogous derivations of Lemma 11 in Petrik et al. (2016) (for the case of finite-horizon MDPs), for the case of finite-horizon MDPs, one has the following chain of inequalities for any given control sequence $\{u_t\}_{t=0}^{T-1}$ and initial observation $x_0$:

$$
\begin{aligned}
&|L(U, \widehat{P}, x_0) - L(U, P, x_0)| \\
&= \left| \mathbb{E}\left[ c_T(x_T) + \sum_{t=0}^{T-1} c_t(x_t, u_t) \,|\, \widehat{P}, x_0 \right] - \mathbb{E}\left[ c_T(x_T) + \sum_{t=0}^{T-1} c_t(x_t, u_t) \,|\, P, x_0 \right] \right| \\
&\leq T^2 \cdot c_{\max} \mathbb{E}\left[ \frac{1}{T} \sum_{t=0}^{T-1} D_{\mathrm{TV}}(P(\cdot|x_t, u_t)||\widehat{P}(\cdot|x_t, u_t)) \,|\, P, x_0 \right] \\
&\leq \sqrt{2} T^2 \cdot c_{\max} \mathbb{E}\left[ \frac{1}{T} \sum_{t=0}^{T-1} \sqrt{\mathrm{KL}(P(\cdot|x_t, u_t)||\widehat{P}(\cdot|x_t, u_t))} \,|\, P, x_0 \right] \\
&\leq \sqrt{2} T^2 \cdot c_{\max} \sqrt{ \mathbb{E}\left[ \frac{1}{T} \sum_{t=0}^{T-1} \mathrm{KL}(P(\cdot|x_t, u_t)||\widehat{P}(\cdot|x_t, u_t)) \,|\, P, x_0 \right]},
\end{aligned}
$$

where $D_{\mathrm{TV}}$ is the total variation distance of two distributions. The first inequality is based on the result of the above lemma, the second inequality is based on Pinsker's inequality (Ordentlich & Weinberger, 2005), and the third inequality is based on Jensen's inequality (Boyd & Vandenberghe, 2004) of $\sqrt{(\cdot)}$ function.

Now consider the expected cumulative KL cost: $\mathbb{E}\left[ \frac{1}{T} \sum_{t=0}^{T-1} \mathrm{KL}(P(\cdot|x_t, u_t)||\widehat{P}(\cdot|x_t, u_t)) \,|\, P, x_0 \right]$ with respect to some arbitrary control action sequence $\{u_t\}_{t=0}^{T-1}$. Notice that this arbitrary action sequence can always be expressed in form of deterministic policy $u_t = \pi'(x_t, t)$ with some non-stationary state-action mapping $\pi'$. Therefore, this KL cost can be written as:

$$
\begin{aligned}
&\mathbb{E}\left[ \frac{1}{T} \sum_{t=0}^{T-1} \mathrm{KL}(P(\cdot|x_t, u_t)||\widehat{P}(\cdot|x_t, u_t)) \,|\, P, \pi, x_0 \right] \\
&= \mathbb{E}\left[ \frac{1}{T} \sum_{t=0}^{T-1} \int_{u_t \in \mathcal{U}} \mathrm{KL}(P(\cdot|x_t, u_t)||\widehat{P}(\cdot|x_t, u_t)) d\pi'(u_t|x_t, t) \,|\, P, x_0 \right] \\
&= \mathbb{E}\left[ \frac{1}{T} \sum_{t=0}^{T-1} \int_{u_t \in \mathcal{U}} \mathrm{KL}(P(\cdot|x_t, u_t)||\widehat{P}(\cdot|x_t, u_t)) \cdot \frac{d\pi'(u_t|x_t, t)}{dU(u_t)} \cdot dU(u_t) \,|\, P, x_0 \right] \\
&\leq \overline{U} \cdot \mathbb{E}_{x,u}\left[ \mathrm{KL}(P(\cdot|x, u)||\widehat{P}(\cdot|x, u)) \right],
\end{aligned}
\tag{10}
$$

where the expectation is taken over the state-action occupation measure $\frac{1}{T} \sum_{t=0}^{T-1} \mathbb{P}(x_t = x, u_t = u|x_0, U)$ of the finite-horizon problem that is induced by data-sampling policy $U$. The last inequality is due to change of measures in policy, and the last inequality is due to the facts that (i) $\pi$ is a deterministic policy, (ii) $dU(u_t)$ is a sampling policy with lebesgue measure $1/\overline{U}$ over all control actions, (iii) the following bounds for importance sampling factor holds: $\left| \frac{d\pi'(u_t|x_t, t)}{dU(u_t)} \right| \leq \overline{U}$.

To conclude the first part of the proof, combining all the above arguments we have the following inequality for any model $\widehat{P}$ and control sequence $U$:

$$
|L(U, \widehat{P}, x_0) - L(U, P, x_0)| \leq \sqrt{2} T^2 \cdot c_{\max} \overline{U} \cdot \sqrt{ \mathbb{E}_{x,u}\left[ \mathrm{KL}(P(\cdot|x, u)||\widehat{P}(\cdot|x, u)) \right]}.
\tag{11}
$$

For the second part of the proof, consider the solution of (SOC3), namely $(U_3^*, \widehat{P}_3^*)$. Using the optimality condition of this problem one obtains the following inequality:

$$
\begin{aligned}
&L(U_3^*, \widehat{P}_3^*, x_0) + \sqrt{2}T^2 \cdot c_{\max}\overline{U} \cdot \sqrt{\mathbb{E}_{x,u}\left[\mathrm{KL}(P(\cdot|x,u)||\widehat{P}_3^*(\cdot|x,u))\right]} \\
&\leq L(U_1^*, \widehat{P}_3^*, x_0) + \sqrt{2}T^2 \cdot c_{\max}\overline{U} \cdot \sqrt{\mathbb{E}_{x,u}\left[\mathrm{KL}(P(\cdot|x,u)||\widehat{P}_3^*(\cdot|x,u))\right]}.
\end{aligned}
\tag{12}
$$

Using the results in (11) and (12), one can then show the following chain of inequalities:

$$
\begin{aligned}
L(U_1^*, P, c, x_0) \geq & L(U_1^*, \widehat{P}_3^*, c, x_0) - \sqrt{2}T^2 \cdot c_{\max}\overline{U} \cdot \sqrt{\mathbb{E}_{x,u}\left[\mathrm{KL}(P(\cdot|x,u)||\widehat{P}_3^*(\cdot|x,u))\right]} \\
= & L(U_1^*, \widehat{P}_3^*, c, x_0) + \sqrt{2}T^2 \cdot c_{\max}\overline{U} \cdot \sqrt{\mathbb{E}_{x,u}\left[\mathrm{KL}(P(\cdot|x,u)||\widehat{P}_3^*(\cdot|x,u))\right]} \\
& - 2\sqrt{2}T^2 \cdot c_{\max}\overline{U} \cdot \sqrt{\mathbb{E}_{x,u}\left[\mathrm{KL}(P(\cdot|x,u)||\widehat{P}_3^*(\cdot|x,u))\right]} \\
\geq & L(U_3^*, \widehat{P}_3^*, c, x_0) + \sqrt{2}T^2 \cdot c_{\max}\overline{U} \cdot \sqrt{\mathbb{E}_{x,u}\left[\mathrm{KL}(P(\cdot|x,u)||\widehat{P}_3^*(\cdot|x,u))\right]} \\
& - 2\sqrt{2}T^2 \cdot c_{\max}\overline{U} \cdot \sqrt{\mathbb{E}_{x,u}\left[\mathrm{KL}(P(\cdot|x,u)||\widehat{P}_3^*(\cdot|x,u))\right]} \\
\geq & L(U_3^*, P, c, x_0) - 2\sqrt{2}T^2 \cdot c_{\max}\overline{U} \cdot \sqrt{\mathbb{E}_{x,u}\left[\mathrm{KL}(P(\cdot|x,u)||\widehat{P}_3^*(\cdot|x,u))\right]},
\end{aligned}
\tag{13}
$$

where $U_1^*$ is the optimizer of (SOC1) and $(U_3^*, \widehat{P}_3^*)$ is the optimizer of (SOC3).

Therefore by letting $\lambda_3 = \sqrt{2}T^2 \cdot c_{\max}\overline{U}$ and $R_3(\widehat{P}) = \mathbb{E}_{x,u}\left[\mathrm{KL}(P(\cdot|x,u)||\widehat{P}(\cdot|x,u))\right]$ and by combining all of the above arguments, the proof of the above lemma is completed.

### A.2 PROOF OF LEMMA 2

For the first part of the proof, at any time-step $t \geq 1$, for any arbitrary control action sequence $\{u_t\}_{t=0}^{T-1}$, and any model $\widehat{P}$, consider the following decomposition of the expected cost :

$$
\begin{aligned}
\mathbb{E}[c(x_t, u_t) \mid \widehat{P}, x_0] = & \int_{x_{0:t-1}\in\mathcal{X}^t} \prod_{k=1}^{t-1} d\widehat{P}(x_k|x_{k-1}, u_{k-1}) \cdot \\
& \int_{z_t\in\mathcal{Z}} \underbrace{\int_{z_{t-1}'\in\mathcal{Z}} dE(z_{t-1}'|x_{t-1})F(z_t|z_{t-1}', u_{t-1})}_{dG(z_t|x_{t-1}, u_{t-1})} \underbrace{\int_{x_t\in\mathcal{X}} dD(x_t|z_t)c(x_t, u_t)}_{\bar{c}(z_t, u_t)}.
\end{aligned}
$$

Now consider the following cost function: $\mathbb{E}[c(x_{t-1}, u_{t-1}) + c(x_t, u_t) \mid \widehat{P}, x_0]$ for $t > 2$. Using the above arguments, one can express this cost as

$$
\begin{aligned}
& \mathbb{E}[c(x_{t-1}, u_{t-1}) + c(x_t, u_t) \mid \widehat{P}, x_0] \\
= & \int_{x_{0:t-2}\in\mathcal{X}^{t-1}} \prod_{k=1}^{t-2} d\widehat{P}(x_k|x_{k-1}, u_{k-1}) \cdot \int_{z_{t-2}'\in\mathcal{Z}} dE(z_{t-2}'|x_{t-2}) \cdot \int_{z_{t-1}\in\mathcal{Z}} dF(z_{t-1}|z_{t-2}', u_{t-2}) \\
& \left( \bar{c}(z_{t-1}, u_{t-1}) + \int_{x_{t-1}\in\mathcal{X}} dD(x_{t-1}|z_{t-1}) \int_{z_{t-1}', z_t\in\mathcal{Z}} dE(z_{t-1}'|x_{t-1})dF(z_t|z_{t-1}', u_{t-1})\bar{c}(z_t, u_t) \right) \\
\leq & \int_{x_{0:t-2}\in\mathcal{X}^{t-1}} \prod_{k=1}^{t-2} d\widehat{P}(x_k|x_{k-1}, u_{k-1}) \cdot \int_{z_{t-2}\in\mathcal{Z}} dE(z_{t-2}|x_{t-2}) \cdot \\
& \int_{z_{t-1}} dF(z_{t-1}|z_{t-2}, u_{t-2}) \left( \bar{c}(z_{t-1}, u_{t-1}) + \int_{z_t\in\mathcal{Z}} dF(z_t|z_{t-1}, u_{t-1})\bar{c}(z_t, u_t) \right) \\
& + c_{\max} \cdot \int_{x_{0:t-2}\in\mathcal{X}^{t-1}} \prod_{k=1}^{t-2} dP(x_k|x_{k-1}, u_{k-1}) \cdot D_{\mathrm{TV}}\left( \int_{x'\in\mathcal{X}} d\widehat{P}(x'|x_{t-2}, u_{t-2})E(\cdot|x') || \int_{z\in\mathcal{Z}} dE(z|x_{t-2})F(\cdot|z, u_{t-2}) \right)
\end{aligned}
$$

By continuing the above expansion, one can show that

$$\left| \mathbb{E}\left[L(U, F, \overline{c}, z_0) \mid E, x_0\right] - L(U, \widehat{P}, c, x_0) \right|$$

$$\leq T^2 \cdot c_{\max} \mathbb{E}\left[ \frac{1}{T} \sum_{t=0}^{T-1} D_{\text{TV}}((E \circ \widehat{P})(\cdot|x_t, u_t)||(F \circ E)(\cdot|x_t, u_t)) \mid P, x_0 \right]$$

$$\leq \sqrt{2}T^2 \cdot c_{\max} \mathbb{E}\left[ \frac{1}{T} \sum_{t=0}^{T-1} \sqrt{\text{KL}((E \circ \widehat{P})(\cdot|x_t, u_t)||(F \circ E)(\cdot|x_t, u_t))} \mid P, x_0 \right]$$

$$\leq \sqrt{2}T^2 \cdot c_{\max} \sqrt{\mathbb{E}\left[ \frac{1}{T} \sum_{t=0}^{T-1} \text{KL}((E \circ \widehat{P})(\cdot|x_t, u_t)||(F \circ E)(\cdot|x_t, u_t)) \mid P, x_0 \right]},$$

where the last inequality is based on Jensen's inequality of $\sqrt{(\cdot)}$ function.

For the second part of the proof, following similar arguments as in the second part of the proof of Lemma 1, one can show the following chain of inequalities for solution of (SOC3) and (SOC2):

$$L(U_3^*, \widehat{P}_3^*, c, x_0)$$

$$\geq \mathbb{E}\left[L(U_3^*, F_3^*, \overline{c}, z_0) \mid E_3^*, x_0\right] - \sqrt{2}T^2 \cdot c_{\max}\overline{U} \cdot \sqrt{\mathbb{E}_{x,u}\left[\text{KL}((E_3^* \circ \widehat{P}_3^*)(\cdot|x, u)||(F_3^* \circ E_3^*)(\cdot|x, u))\right]}$$

$$= \mathbb{E}\left[L(U_3^*, F_3^*, \overline{c}, z_0) \mid E_3^*, x_0\right] + \sqrt{2}T^2 \cdot c_{\max}\overline{U} \cdot \sqrt{\mathbb{E}_{x,u}\left[\text{KL}((E_3^* \circ \widehat{P}_3^*)(\cdot|x, u)||(F_3^* \circ E_3^*)(\cdot|x, u))\right]}$$

$$- 2\sqrt{2}T^2 \cdot c_{\max}\overline{U} \cdot \sqrt{\mathbb{E}_{x,u}\left[\text{KL}((E_3^* \circ \widehat{P}_3^*)(\cdot|x, u)||(F_3^* \circ E_3^*)(\cdot|x, u))\right]}$$

$$\geq \mathbb{E}\left[L(U_2^*, F_2^*, \overline{c}, z_0) \mid E_2^*, x_0\right] + \sqrt{2}T^2 \cdot c_{\max}\overline{U} \cdot \sqrt{\mathbb{E}_{x,u}\left[\text{KL}((E_2^* \circ \widehat{P}_2^*)(\cdot|x, u)||(F_2^* \circ E_2^*)(\cdot|x, u))\right]}$$

$$- 2\sqrt{2}T^2 \cdot c_{\max}\overline{U} \cdot \sqrt{\mathbb{E}_{x,u}\left[\text{KL}((E_3^* \circ \widehat{P}_3^*)(\cdot|x, u)||(F_3^* \circ E_3^*)(\cdot|x, u))\right]}$$

$$\geq L(U_2^*, \widehat{P}_2^*, c, x_0) - 2\underbrace{\sqrt{2}T^2 \cdot c_{\max}\overline{U}}_{\lambda_2} \cdot \underbrace{\sqrt{\mathbb{E}_{x,u}\left[\text{KL}((E_2^* \circ \widehat{P}_2^*)(\cdot|x, u)||(F_2^* \circ E_2^*)(\cdot|x, u))\right]}}_{R_2''(\widehat{P}_2^*)},$$

$$(14)$$

where the first and third inequalities are based on the first part of this Lemma, and the second inequality is based on the optimality condition of problem (SOC2). This completes the proof.

### A.3 PROOF OF COROLLARY 1

To start with, the total-variation distance $D_{\text{TV}}\left(\int_{x' \in \mathcal{X}} d\widehat{P}(x'|x, u)E(\cdot|x')||(F \circ E)(\cdot|x, u)\right)$ can be bounded by the following inequality using triangle inequality:

$$D_{\text{TV}}\left(\int_{x' \in \mathcal{X}} d\widehat{P}(x'|x, u)E(\cdot|x')||(F \circ E)(\cdot|x, u)\right)$$

$$\leq D_{\text{TV}}\left(\int_{x' \in \mathcal{X}} dP(x'|x, u)E(\cdot|x')||(F \circ E)(\cdot|x, u)\right) + D_{\text{TV}}\left(\int_{x' \in \mathcal{X}} dP(x'|x, u)E(\cdot|x')|| \int_{x' \in \mathcal{X}} d\widehat{P}(x'|x, u)E(\cdot|x')\right)$$

$$\leq D_{\text{TV}}\left(\int_{x' \in \mathcal{X}} dP(x'|x, u)E(\cdot|x')||(F \circ E)(\cdot|x, u)\right) + D_{\text{TV}}\left(P(\cdot|x, u)||\widehat{P}(\cdot|x, u)\right)$$

where the second inequality follows from the convexity property of the $D_{\text{TV}}$-norm (w.r.t. convex weights $E(\cdot|x'), \forall x'$). Then by Pinsker's inequality, one obtains the following inequality:

$$D_{\text{TV}}\left(\int_{x' \in \mathcal{X}} d\widehat{P}(x'|x, u)E(\cdot|x')||(F \circ E)(\cdot|x, u)\right)$$

$$\leq \sqrt{2\text{KL}\left(\int_{x' \in \mathcal{X}} dP(x'|x, u)E(\cdot|x')||(F \circ E)(\cdot|x, u)\right)} + \sqrt{2\text{KL}\left(P(\cdot|x, u)||\widehat{P}(\cdot|x, u)\right)}.$$

$$(15)$$

We now analyze the batch consistency regularizer:

$$R_2''(\widehat{P}) = \mathbb{E}_{x,u,x'}\left[\text{KL}(E(\cdot|x')||(F \circ E)(\cdot|x,u))\right]$$

and connect it with the inequality in (15). Using Jensen's inequality of convex function $x \log x$, for any observation-action pair $(x,u)$ sampled from $U_\tau$, one can show that

$$\int_{x' \in \mathcal{X}} dP(x'|x,u) \int_{z' \in \mathcal{Z}} dE(z'|x') \log \left(\int_{x' \in \mathcal{X}} dP(x'|x,u)E(z'|x')\right)$$
$$\leq \int_{x' \in \mathcal{X}} dP(x'|x,u) \int_{z' \in \mathcal{Z}} dE(z'|x') \log \left(E(z'|x')\right). \tag{16}$$

Therefore, for any observation-control pair $(x,u)$ the following inequality holds:

$$\text{KL} \left(\int_{x' \in \mathcal{X}} dP(x'|x,u)E(\cdot|x')||(F \circ E)(\cdot|x,u)\right)$$
$$= \int_{x' \in \mathcal{X}} dP(x'|x,u) \int_{z' \in \mathcal{Z}} dE(z'|x') \log \left(\int_{x' \in \mathcal{X}} dP(x'|x,u)E(z'|x')\right)$$
$$\quad - \int_{x' \in \mathcal{X}} dP(x'|x,u) \log \left(g(x'|x,u)\right) \tag{17}$$
$$\leq \int_{x' \in \mathcal{X}} dP(x'|x,u) \int_{z' \in \mathcal{Z}} dE(z'|x') \log \left(E(z'|x')\right) - \int_{x' \in \mathcal{X}} dP(x'|x,u) \log \left(g(x'|x,u)\right)$$
$$= \text{KL}(E(\cdot|x')||(F \circ E)(\cdot|x,u))$$

By taking expectation over $(x,u)$ one can show that

$$\mathbb{E}_{x,u}\left[\text{KL}(\int_{x' \in \mathcal{X}} dP(x'|x,u)E(\cdot|x')||(F \circ E)(\cdot|x,u))\right]$$

is the lower bound of the batch consistency regularizer. Therefore, the above arguments imply that

$$D_{\text{TV}}\left(\int_{x' \in \mathcal{X}} d\widehat{P}(x'|x,u)E(\cdot|x')||(F \circ E)(\cdot|x,u)\right) \leq \sqrt{2}\sqrt{R_2''(\widehat{P}) + R_3(\widehat{P})}. \tag{18}$$

The inequality is based on the property that $\sqrt{a} + \sqrt{b} \leq \sqrt{2}\sqrt{a+b}$.

Equipped with the above additional results, the rest of the proof on the performance bound follows directly from the results from Lemma 2, in which here we further upper-bound $D_{\text{TV}}\left(\int_{x' \in \mathcal{X}} d\widehat{P}(x'|x,u)E(\cdot|x')||(F \circ E)(\cdot|x,u)\right)$, when $\widehat{P} = \widehat{P}_2^*$.

## A.4 Proof of Lemma 3

For the first part of the proof, at any time-step $t \geq 1$, for any arbitrary control action sequence $\{u_t\}_{t=0}^{T-1}$ and for any model $\widehat{P}$, consider the following decomposition of the expected cost :

$$\mathbb{E}[c(x_t,u_t) \mid P, x_0] = c_{\max} \cdot \int_{x_{0:t-1} \in \mathcal{X}^t} \prod_{k=1}^{t-1} dP(x_k|x_{k-1},u_{k-1})D_{\text{TV}}(P(\cdot|x_{t-1},u_{t-1})||\widehat{P}(\cdot|x_{t-1},u_{t-1}))$$

$$+ \int_{x_{0:t-1} \in \mathcal{X}^t} \prod_{k=1}^{t-1} dP(x_k|x_{k-1},u_{k-1}) \int_{z_t \in \mathcal{Z}} \underbrace{\int_{z'_{t-1} \in \mathcal{Z}} dE(z'_{t-1}|x_{t-1})F(z_t|z'_{t-1},u_{t-1})}_{dG(z_t|x_{t-1},u_{t-1})} \underbrace{\int_{x_t \in \mathcal{X}} dD(x_t|z_t)c(x_t,u_t)}_{\bar{c}(z_t,u_t)}.$$

Now consider the following cost function: $\mathbb{E}[c(x_{t-1}, u_{t-1}) + c(x_t, u_t) \mid \widehat{P}, x_0]$ for $t > 2$. Using the above arguments, one can express this cost as

$$\mathbb{E}[c(x_{t-1}, u_{t-1}) + c(x_t, u_t) \mid P, x_0]$$

$$= \int_{x_{0:t-2} \in \mathcal{X}^{t-1}} \prod_{k=1}^{t-2} dP(x_k | x_{k-1}, u_{k-1}) \cdot \int_{z'_{t-2} \in \mathcal{Z}} dE(z'_{t-2} | x_{t-2}) \cdot \int_{z_{t-1}} dF(z_{t-1} | z'_{t-2}, u_{t-2}) \cdot$$

$$\left( \bar{c}(z_{t-1}, u_{t-1}) + \int_{x_{t-1}} dD(x_{t-1}|z_{t-1}) \int_{z'_{t-1}, z_t \in \mathcal{Z}} dE(z'_{t-1}|x_{t-1}) dF(z_t|z'_{t-1}, u_{t-1}) \bar{c}(z_t, u_t) \right)$$

$$+ c_{\max} \cdot \sum_{j=1}^{2} j \cdot \int_{x_{0:t-j}} \prod_{k=1}^{t-j} dP(x_k|x_{k-1}, u_{k-1}) D_{\text{TV}}(P(\cdot|x_{t-j}, u_{t-j}) || \widehat{P}(\cdot|x_{t-j}, u_{t-j}))$$

$$\leq \int_{x_{0:t-2} \in \mathcal{X}^{t-1}} \prod_{k=1}^{t-2} dP(x_k|x_{k-1}, u_{k-1}) \cdot \int_{z_{t-2} \in \mathcal{Z}} dE(z_{t-2}|x_{t-2}) \cdot$$

$$\int_{z_{t-1}} dF(z_{t-1}|z_{t-2}, u_{t-2}) \left( \bar{c}(z_{t-1}, u_{t-1}) + \int_{z_t \in \mathcal{Z}} dF(z_t|z_{t-1}, u_{t-1}) \bar{c}(z_t, u_t) \right)$$

$$+ c_{\max} \cdot \sum_{j=1}^{2} j \cdot \int_{x_{0:t-j}} \prod_{k=1}^{t-j} dP(x_k|x_{k-1}, u_{k-1}) D_{\text{TV}}(P(\cdot|x_{t-j}, u_{t-j}) || \widehat{P}(\cdot|x_{t-j}, u_{t-j}))$$

$$+ c_{\max} \cdot \int_{x_{0:t-2} \in \mathcal{X}^{t-1}} \prod_{k=1}^{t-2} dP(x_k|x_{k-1}, u_{k-1}) \cdot D_{\text{TV}} \left( \int_{x' \in \mathcal{X}} d\widehat{P}(x'|x_{t-2}, u_{t-2}) E(\cdot|x') || \int_{z \in \mathcal{Z}} dE(z|x_{t-2}) F(\cdot|z, u_{t-2}) \right).$$

Continuing the above expansion, one can show that

$$|\mathbb{E}[L(U, F, \bar{c}, z_0) \mid E, x_0] - L(U, P, x_0)|$$

$$\leq T^2 \cdot c_{\max} \mathbb{E} \left[ \frac{1}{T} \sum_{t=0}^{T-1} D_{\text{TV}}(P(\cdot|x_t, u_t)||\widehat{P}(\cdot|x_t, u_t)) + D_{\text{TV}}(\int_{x' \in \mathcal{X}} d\widehat{P}(x'|x_t, u_t) E(\cdot|x')||(F \circ E)(\cdot|x_t, u_t)) \mid P, x_0 \right]$$

$$\leq \sqrt{2} T^2 \cdot c_{\max} \mathbb{E} \left[ \frac{1}{T} \sum_{t=0}^{T-1} \sqrt{\text{KL}(P(\cdot|x_t, u_t)||\widehat{P}(\cdot|x_t, u_t))} + \sqrt{\text{KL}(\int_{x' \in \mathcal{X}} d\widehat{P}(x'|x_t, u_t) E(\cdot|x')||(F \circ E)(\cdot|x_t, u_t))} \mid P, x_0 \right]$$

$$\leq \sqrt{2} T^2 \cdot c_{\max} \mathbb{E} \left[ \frac{1}{T} \sum_{t=0}^{T-1} \sqrt{\text{KL}(P(\cdot|x_t, u_t)||\widehat{P}(\cdot|x_t, u_t))} \right.$$

$$\left. + \sqrt{\text{KL}(P(\cdot|x_t, u_t)||\widehat{P}(\cdot|x_t, u_t)) + \text{KL}(E(\cdot|x_{t+1})||(F \circ E)(\cdot|x_t, u_t))} \mid P, x_0 \right]$$

$$\leq 2T^2 \cdot c_{\max} \sqrt{\mathbb{E} \left[ \frac{1}{T} \sum_{t=0}^{T-1} 3\text{KL}(P(\cdot|x_t, u_t)||\widehat{P}(\cdot|x_t, u_t)) + 2\text{KL}(E(\cdot|x_{t+1})||(F \circ E)(\cdot|x_t, u_t)) \mid P, x_0 \right]},$$

where the last inequality is based on the fact that $\sqrt{a} + \sqrt{b} \leq \sqrt{2}\sqrt{a+b}$ and is based on Jensen's inequality of $\sqrt{(\cdot)}$ function.

For the second part of the proof, following similar arguments from Lemma 2, one can show the following inequality for the solution of (SOC3) and (SOC2):

$$L(U_1^*, P, c, x_0) \geq \mathbb{E}\left[L(U_1^*, F_2^*, \bar{c}, z_0) \mid E_2^*, x_0\right] - \sqrt{2}T^2 \cdot c_{\max}\overline{U} \cdot \sqrt{2R_2''(\widehat{P}_2^*) + 3R_3(\widehat{P}_2^*)}$$

$$= \mathbb{E}\left[L(U_1^*, F_2^*, \bar{c}, z_0) \mid E_2^*, x_0\right] + \sqrt{2}T^2 \cdot c_{\max}\overline{U} \cdot \sqrt{2R_2''(\widehat{P}_2^*) + 3R_3(\widehat{P}_2^*)}$$

$$- 2\sqrt{2}T^2 \cdot c_{\max}\overline{U} \cdot \sqrt{2R_2''(\widehat{P}_2^*) + 3R_3(\widehat{P}_2^*)}$$

$$\geq \mathbb{E}\left[L(U_2^*, F_2^*, \bar{c}, z_0) \mid E_2^*, x_0\right] + \sqrt{2}T^2 \cdot c_{\max}\overline{U} \cdot \sqrt{2R_2''(\widehat{P}_2^*) + 3R_3(\widehat{P}_2^*)} \quad (19)$$

$$- 2\sqrt{2}T^2 \cdot c_{\max}\overline{U} \cdot \sqrt{2R_2''(\widehat{P}_2^*) + 3R_3(\widehat{P}_2^*)}$$

$$\geq L(U_2^*, P, c, x_0) - 2 \underbrace{\sqrt{2}T^2 \cdot c_{\max}\overline{U}}_{\lambda_2} \cdot \sqrt{2R_2''(\widehat{P}_2^*) + 3R_3(\widehat{P}_2^*)},$$

where the first and third inequalities are based on the first part of this Lemma, and the second inequality is based on the optimality condition of problem (SOC2). This completes the proof.

### A.5 PROOF OF LEMMA 4

**A Recap of the Result:** Let $(U^*_{\text{LLC}}, \widehat{P}^*_{\text{LLC}})$ be a LLC solution to (SOC-LLC) and $U^*_1$ be a solution to (SOC1). Suppose the nominal latent state-action pair $\{(\boldsymbol{z}_t, \boldsymbol{u}_t)\}_{t=0}^{T-1}$ satisfies the condition: $(\boldsymbol{z}_t, \boldsymbol{u}_t) \sim \mathcal{N}((z^*_{2,t}, u^*_{2,t}), \delta^2 I)$, where $\{(z^*_{2,t}, u^*_{2,t}\}_{t=0}^{T-1}$ is the optimal trajectory of problem (SOC2). Then with probability $1 - \eta$, we have $L(U^*_1, P, c, x_0) \geq L(U^*_{\text{LLC}}, P, c, x_0) - 2\lambda_{\text{LLC}}\sqrt{R_2(\widehat{P}^*_{\text{LLC}}) + R_{\text{LLC}}(\widehat{P}^*_{\text{LLC}})}$.

**Discussions of the effect of $\delta$ on LLC Performance:** The result of this lemma shows that when the nominal state and actions are $\delta$-*close* to the optimal trajectory of (SOC2), i.e., at each time step $(\boldsymbol{z}_t, \boldsymbol{u}_t)$ is a sample from the Gaussian distribution centered at $(z^*_{2,t}, u^*_{2,t})$ with standard deviation $\delta$, then one can obtain a performance bound of LLC algorithm that is in terms of the regularization loss $R_{\text{LLC}}$. To quantify the above condition, one can use Mahalanobis distance (De Maesschalck et al., 2000) to measure the distance of $(\boldsymbol{z}_t, \boldsymbol{u}_t)$ to distribution $\mathcal{N}((z^*_{2,t}, u^*_{2,t}), \delta^2 I)$, i.e., we want to check for the condition:

$$\frac{\|(\boldsymbol{z}_t, \boldsymbol{u}_t) - (z^*_{2,t}, u^*_{2,t})\|}{\delta} \leq \epsilon', \ \forall t,$$

for any arbitrary error tolerance $\epsilon' > 0$. While we cannot verify the condition without knowing the optimal trajectory $\{(z^*_{2,t}, u^*_{2,t})\}_{t=0}^{T-1}$, the above condition still offers some insights in choosing the parameter $\delta$ based on the trade-off of designing nominal trajectory $\{(\boldsymbol{z}_t, \boldsymbol{u}_t)\}_{t=0}^{T-1}$ and optimizing $R_{\text{LLC}}$. When $\delta$ is large, the low-curvature regularization imposed by the $R_{\text{LLC}}$ regularizer will cover a large portion of the state-action space. In the extreme case when $\delta \to \infty$, $R_{\text{LLC}}$ can be viewed as a regularizer that enforces *global linearity*. Here the trade-off is that the loss $R_{\text{LLC}}$ is generally higher, which in turn degrades the performance bound of the LLC control algorithm in Lemma 4. On the other hand, when $\delta$ is small the low-curvature regularization in $R_{\text{LLC}}$ only covers a smaller region of the latent state-action space, and thus the loss associated with this term is generally lower (which provides a tighter performance bound in Lemma 4). However the performance result will only hold when $(\boldsymbol{z}_t, \boldsymbol{u}_t)$ happens to be close to $(z^*_{2,t}, u^*_{2,t})$ at each time-step $t \in \{0, \ldots, T-1\}$.

**Proof:** For simplicity, we will focus on analyzing the noiseless case when the dynamics is deterministic (i.e., $\Sigma_w = 0$). Extending the following analysis for the case of non-deterministic dynamics should be straight-forward.

First, consider any arbitrary latent state-action pair $(z, u)$, such that the corresponding nominal state-action pair $(\boldsymbol{z}, \boldsymbol{u})$ is constructed by $\boldsymbol{z} = z - \delta z, \boldsymbol{u} = u - \delta u$, where $(\delta z, \delta u)$ is sampled from the Gaussian distribution $\mathcal{N}(0, \delta^2 I)$. (The random vectors are denoted as $(\delta z', \delta u')$) By the two-tailed Bernstein's inequality (Murphy, 2012), for any arbitrarily given $\eta \in (0, 1]$ one has the following inequality with probability $1 - \eta$:

$$|f_{\mathcal{Z}}(\boldsymbol{z}, \boldsymbol{u}) + A(\boldsymbol{z}, \boldsymbol{u})\delta z + B(\boldsymbol{z}, \boldsymbol{u})\delta u - f_{\mathcal{Z}}(z, u)|$$
$$\leq \sqrt{2\log(2/\eta)}\sqrt{\mathbb{V}_{(\delta z', \delta u') \sim \mathcal{N}(0, \delta^2 I)}[f_{\mathcal{Z}}(\boldsymbol{z}, \boldsymbol{u}) + A(\boldsymbol{z}, \boldsymbol{u})\delta z' + B(\boldsymbol{z}, \boldsymbol{u})\delta u' - f_{\mathcal{Z}}(z, u)]}$$
$$+ \left|\mathbb{E}_{(\delta z', \delta u') \sim \mathcal{N}(0, \delta^2 I)}[f_{\mathcal{Z}}(\boldsymbol{z}, \boldsymbol{u}) + A(\boldsymbol{z}, \boldsymbol{u})\delta z' + B(\boldsymbol{z}, \boldsymbol{u})\delta u' - f_{\mathcal{Z}}(z, u)]\right|$$
$$\leq (1 + \sqrt{2\log(2/\eta)})\bigg(\underbrace{\mathbb{E}_{(\delta z', \delta u') \sim \mathcal{N}(0, \delta^2 I)}\big[\|f_{\mathcal{Z}}(\boldsymbol{z}, \boldsymbol{u}) + A(\boldsymbol{z}, \boldsymbol{u})\delta z' + B(\boldsymbol{z}, \boldsymbol{u})\delta u' - f_{\mathcal{Z}}(z, u)\|^2\big]}_{R_{\text{LLC}}(\widehat{P}|z, u)}\bigg)^{1/2}.$$

The second inequality is due to the basic fact that variance is less than second-order moment of a random variable. On the other hand, at each time step $t \in \{0, \ldots, T-1\}$ by the Lipschitz property of the immediate cost, the value function $V_t(z) = \min_{U_{t:T-1}} \mathbb{E}\left[\bar{c}_T(z_T) + \sum_{\tau=t}^{T-1} \bar{c}_\tau(z_\tau, u_\tau) \mid z_t = z\right]$ is also Lipchitz with constant $(T - t + 1)c_{\text{lip}}$. Using the Lipschitz property of $V_{t+1}$, for any $(z, u)$

and $(\delta z, \delta u)$, such that $(\boldsymbol{z}, \boldsymbol{u}) = (z - \delta z, u - \delta u)$, one has the following property:

$$
\begin{aligned}
&|V_{t+1}(\boldsymbol{z}' + A(\boldsymbol{z}, \boldsymbol{u})\delta z + B(\boldsymbol{z}, \boldsymbol{u})\delta u) - V_{t+1}(f_{\mathcal{Z}}(z, u))| \\
&\leq (T - t)c_{\text{lip}} \cdot |f_{\mathcal{Z}}(\boldsymbol{z}, \boldsymbol{u}) + A(\boldsymbol{z}, \boldsymbol{u})\delta z + B(\boldsymbol{z}, \boldsymbol{u})\delta u - f_{\mathcal{Z}}(z, u)|,
\end{aligned}
\tag{20}
$$

Therefore, at any arbitrary state-action pair $(\tilde{z}, \tilde{u})$, for $\boldsymbol{z} = z - \delta z$, and $\boldsymbol{u} = \tilde{u} - \delta u$ with Gaussian sample $(\delta z, \delta u) \sim \mathcal{N}(0, \delta^2 I)$, the following inequality on the value function holds w.p. $1 - \eta$:

$$
V_{t+1}(f_{\mathcal{Z}}(\tilde{z}, \tilde{u})) \geq V_{t+1}(\boldsymbol{z}' + A(\boldsymbol{z}, \boldsymbol{u})\delta z + B(\boldsymbol{z}, \boldsymbol{u})\delta u) - (T - t)c_{\text{lip}}(1 + \sqrt{2\log(2/\eta)}) \cdot \sqrt{R_{\text{LLC}}(\widehat{P}|\tilde{z}, \tilde{u})},
$$

which further implies

$$
\begin{aligned}
&\bar{c}_t(\tilde{z}, \tilde{u}) + V_{t+1}(f_{\mathcal{Z}}(\tilde{z}, \tilde{u})) \\
&\geq \bar{c}_t(\tilde{z}, \tilde{u}) + V_{t+1}(\boldsymbol{z}' + A(\boldsymbol{z}, \boldsymbol{u})\delta z + B(\boldsymbol{z}, \boldsymbol{u})\delta u) - (T - t)c_{\text{lip}}(1 + \sqrt{2\log(2/\eta)}) \cdot \sqrt{R_{\text{LLC}}(\widehat{P}|\tilde{z}, \tilde{u})},
\end{aligned}
$$

Now let $\tilde{u}^*$ be the optimal control w.r.t. Bellman operator $T_t[V_{t+1}](\tilde{z})$ at any latent state $\tilde{z}$. Based on the assumption of this lemma, at each state $\tilde{z}$ the nominal latent state-action pair $(\boldsymbol{z}, \boldsymbol{u})$ is generated by perturbing $(\tilde{z}, \tilde{u}^*)$ with Gaussian sample $(\delta z, \delta u) \sim \mathcal{N}(0, \delta^2 I)$ that is in form of $\boldsymbol{z} = \tilde{z} - \delta z$, $\boldsymbol{u} = \tilde{u} - \delta u$. Then by the above arguments the following chain of inequalities holds w.p. $1 - \eta$:

$$
\begin{aligned}
T_t[V_{t+1}](\tilde{z}) :=& \min_{\tilde{u}} \bar{c}_t(\tilde{z}, \tilde{u}) + V_{t+1}(f_{\mathcal{Z}}(\tilde{z}, \tilde{u})) \\
=& \bar{c}_t(\tilde{z}, \tilde{u}^*) + V_{t+1}(f_{\mathcal{Z}}(\tilde{z}, \tilde{u}^*)) \\
\geq& \bar{c}_t(\tilde{z}, \tilde{u}^*) + V_{t+1}(f_{\mathcal{Z}}(\boldsymbol{z}, \boldsymbol{u}) + A(\boldsymbol{z}, \boldsymbol{u})\delta z + B(\boldsymbol{z}, \boldsymbol{u})\delta u) \\
& - |V_{t+1}(\boldsymbol{z}' + A(\boldsymbol{z}, \boldsymbol{u})\delta z + B(\boldsymbol{z}, \boldsymbol{u})\delta u) - V_{t+1}(f_{\mathcal{Z}}(\tilde{z}, \tilde{u}^*))| \\
\geq& \bar{c}_t(\tilde{z}, \boldsymbol{u} + \delta u) + V_{t+1}(f_{\mathcal{Z}}(\boldsymbol{z}, \boldsymbol{u}) + A(\boldsymbol{z}, \boldsymbol{u})\delta z + B(\boldsymbol{z}, \boldsymbol{u})\delta u) \\
& - (T - t)c_{\text{lip}}(1 + \sqrt{2\log(2/\eta)})\sqrt{\max_{z,u} R_{\text{LLC}}(\widehat{P}|z, u)} \\
\geq& \min_{\delta u} \bar{c}_t(\tilde{z}, \boldsymbol{u} + \delta u) + V_{t+1}(f_{\mathcal{Z}}(\boldsymbol{z}, \boldsymbol{u}) + A(\boldsymbol{z}, \boldsymbol{u})\delta z + B(\boldsymbol{z}, \boldsymbol{u})\delta u) \\
& - (T - t)c_{\text{lip}}(1 + \sqrt{2\log(2/\eta)})\sqrt{\max_{z,u} R_{\text{LLC}}(\widehat{P}|z, u)}
\end{aligned}
\tag{21}
$$

Recall the LLC loss function is given by

$$
R_{\text{LLC}}(\widehat{P}) = \mathbb{E}_{x,u}\left[\mathbb{E}\left[R_{\text{LLC}}(\widehat{P}|z, u) \mid z\right] \mid E\right].
$$

Also consider the Bellman operator w.r.t. latent SOC: $T_t[V](z) = \min_u \bar{c}_t(z, u) + V(f_{\mathcal{Z}}(z, u))$, and the Bellman operator w.r.t. LLC: $T_{t,\text{LLC}}[V](z) = \min_{\delta u} \bar{c}_t(z, \delta u + \boldsymbol{u}) + V(f_{\mathcal{Z}}(\boldsymbol{z}, \boldsymbol{u}) + A(\boldsymbol{z}, \boldsymbol{u})\delta z + B(\boldsymbol{z}, \boldsymbol{u})\delta u)$. Utilizing these definitions, the inequality in (21) can be further expressed as

$$
T_t[V_{t+1}](\tilde{z}) \geq T_{t,\text{LLC}}[V_{t+1}](\tilde{z}) - (T - t)c_{\text{lip}}c_{\max}(1 + \sqrt{2\log(2/\eta)})\sqrt{\overline{UX}}\sqrt{R_{\text{LLC}}(\widehat{P})}, \tag{22}
$$

This inequality is due to the fact that all latent states are generated by the encoding observations, i.e., $z \sim E(\cdot|x)$, and thus by following analogous arguments as in the proof of Lemma 1, one has

$$
\max_{z,u} R_{\text{LLC}}(\widehat{P}|z, u) \leq \overline{UX}\mathbb{E}_{x,u}\left[\mathbb{E}\left[R_{\text{LLC}}(\widehat{P}|z, u) \mid z\right] \mid E\right] = \overline{UX}R_{\text{LLC}}(\widehat{P}).
$$

Therefore, based on the dynamic programming result that bounds the difference of value function w.r.t. different Bellman operators in finite-horizon problems (for example see Theorem 1.3 in Bertsekas (1995)), the above inequality implies the following bound in the value function, w.p. $1 - \eta$:

$$
\begin{aligned}
&\min_{U,\widehat{P}} L(U, F, \bar{c}, z_0) \\
&\geq L(U^*_{\text{LLC}}, \widehat{P}^*_{\text{LLC}}, \bar{c}, z_0) - \sum_{t=1}^{T-1}(T - t) \cdot c_{\text{lip}}c_{\max} \cdot T \cdot (1 + \sqrt{2\log(2T/\eta)}) \cdot \sqrt{\overline{UX}} \cdot \sqrt{R_{\text{LLC}}(\widehat{P}^*_{\text{LLC}})} \\
&\geq L(U^*_{\text{LLC}}, \widehat{P}^*_{\text{LLC}}, \bar{c}, z_0) - T^2 \cdot c_{\text{lip}}c_{\max} \cdot (1 + \sqrt{2\log(2T/\eta)}) \cdot \sqrt{\overline{UX}} \cdot \sqrt{R_{\text{LLC}}(\widehat{P}^*_{\text{LLC}})}.
\end{aligned}
\tag{23}
$$

Notice that here we replace $\eta$ in the result in (22) with $\eta/T$. In order to prove (23), we utilize (22) for each $t \in \{0, \ldots, T-1\}$, and this replacement is the result of applying the Union Probability bound (Murphy, 2012) (to ensure (23) holds with probability $1 - \eta$).

Therefore the proof is completed by combining the above result with that in Lemma 3.

## B    THE LATENT SPACE iLQR ALGORITHM

### B.1    PLANNING IN THE LATENT SPACE (HIGH-LEVEL DESCRIPTION)

We follow the same control scheme as in Banijamali et al. (2018). Namely, we use the iLQR (Li & Todorov, 2004) solver to plan in the latent space. Given a start observation $x_{\text{start}}$ and a goal observation $x_{\text{goal}}$, corresponding to underlying states $\{s_{\text{start}}, s_{\text{goal}}\}$, we encode the observations to retrieve $z_{\text{start}}$ and $z_{\text{goal}}$. Then, the procedure goes as follows: we initialize a random trajectory (sequence of actions), feed it to the iLQR solver and apply the first action from the trajectory the solver outputs. We observe the next observation returned from the system (closed-loop control), and feed the updated trajectory to the iLQR solver. This procedure continues until the it reaches the end of the problem horizon. We use a receding window approach, where at every planning step the solver only optimizes for a fixed length of actions sequence, independent of the problem horizon.

### B.2    DETAILS ABOUT iLQR IN THE LATENT SPACE

Consider the latent state SOC problem

$$\min_{U} \mathbb{E}\left[\bar{c}_T(z_T) + \sum_{t=0}^{T-1} \bar{c}_t(z_t, u_t) \mid z_0\right].$$

At each time instance $t \in \{0, \dots, T\}$ the value function of this problem is given by

$$V_T(z) = \bar{c}_T(z), \ V_t(z) = \min_{U_{t:T-1}} \mathbb{E}\left[\bar{c}_T(z_T) + \sum_{\tau=t}^{T-1} \bar{c}_\tau(z_\tau, u_\tau) \mid z_t = z\right], \ \forall t < T. \quad (24)$$

Recall that the nonlinear latent space dynamics model is given by:

$$z_{t+1} = F(z_t, u_t, w_t) := F_\mu(z_t, u_t) + F_\sigma \cdot w_t, \ w_t \sim \mathcal{N}(0, I), \ \forall t \geq 0, \quad (25)$$

where $F_\mu(z_t, u_t)$ is the deterministic dynamics model and $F_\sigma^\top F_\sigma$ is the covariance of the latent dynamics system noise. Notice that the deterministic dynamics model $F_\mu(z_t, u_t)$ is smooth, and therefore the following Jacobian terms are well-posed:

$$A(z, u) := \frac{\partial F_\mu(z, u)}{\partial z}, \ B(z, u) := \frac{\partial F_\mu(z, u)}{\partial u}, \ \forall z \in \mathbb{R}^{n_z}, \ \forall u \in \mathbb{R}^{n_u}.$$

By the Bellman's principle of optimality, at each time instance $t \in \{0, \dots, T-1\}$ the value function is a solution of the recursive fixed point equation

$$V_t(z) = \min_{u} \ Q_t(z, u), \quad (26)$$

where the state-action value function at time-instance $t$ w.r.t. state-action pair $(z_t, u_t) = (z, u)$ is given by

$$Q_t(z, u) = \bar{c}_t(z, u) + \mathbb{E}_{w_t}\left[V_{t+1}(F(z_t, u_t, w_t)) \mid z_t = z, u_t = u\right].$$

In the setting of the iLQR algorithm, assume we have access to a trajectory of latent states and actions that is in form of $\{(\boldsymbol{z}_t, \boldsymbol{u}_t, \boldsymbol{z}_{t+1})\}_{t=0}^{T-1}$. At each iteration, the iLQR algorithm has the following steps:

1. Given a nominal trajectory, find an optimal policy w.r.t. the perturbed latent states
2. Generate a sequence of optimal perturbed actions that locally improves the cumulative cost of the given trajectory
3. Apply the above sequence of actions to the environment and update the nominal trajectory
4. Repeat the above steps with new nominal trajectory

Denote by $\delta z_t = z_t - \boldsymbol{z}_t$ and $\delta u_t = u_t - \boldsymbol{u}_t$ the deviations of state and control action at time step $t$ respectively. Assuming that the nominal next state $\boldsymbol{z}_{t+1}$ is generated by the deterministic transition $F_\mu(\boldsymbol{z}_t, \boldsymbol{u}_t)$ at the nominal state and action pair $(\boldsymbol{z}_t, \boldsymbol{u}_t)$, the first-order Taylor series approximation of the latent space transition is given by

$$\delta z_{t+1} := z_{t+1} - \boldsymbol{z}_{t+1} = A(\boldsymbol{z}_t, \boldsymbol{u}_t)\delta z_t + B(\boldsymbol{z}_t, \boldsymbol{u}_t)\delta u_t + F_\sigma \cdot w_t + O(\|(\delta z_t, \delta u_t)\|^2), \ w_t \sim \mathcal{N}(0, I). \quad (27)$$

To find a locally optimal control action sequence $u_t^* = \pi_{\delta z,t}^*(\delta z_t) + \boldsymbol{u}_t, \forall t$, that improves the cumulative cost of the trajectory, we compute the locally optimal perturbed policy (policy w.r.t. perturbed latent state) $\{\pi_{\delta z,t}^*(\delta z_t)\}_{t=0}^{T-1}$ that minimizes the following second-order Taylor series approximation of $Q_t$ around nominal state-action pair $(\boldsymbol{z}_t, \boldsymbol{u}_t), \forall t \in \{0, \ldots, T-1\}$:

$$Q_t(z_t, u_t) = Q_t(\boldsymbol{z}_t, \boldsymbol{u}_t) + \frac{1}{2}\begin{bmatrix} 1 \\ \delta z_t \\ \delta u_t \end{bmatrix}^\top \begin{bmatrix} F_\sigma^\top F_\sigma & Q_{t,z}(\boldsymbol{z}_t, \boldsymbol{u}_t)^\top & Q_{t,u}(\boldsymbol{z}_t, \boldsymbol{u}_t)^\top \\ Q_{t,z}(\boldsymbol{z}_t, \boldsymbol{u}_t) & Q_{t,zz}(\boldsymbol{z}_t, \boldsymbol{u}_t) & Q_{t,uz}(\boldsymbol{z}_t, \boldsymbol{u}_t)^\top \\ Q_{t,u}(\boldsymbol{z}_t, \boldsymbol{u}_t) & Q_{t,uz}(\boldsymbol{z}_t, \boldsymbol{u}_t) & Q_{t,uu}(\boldsymbol{z}_t, \boldsymbol{u}_t) \end{bmatrix} \begin{bmatrix} 1 \\ \delta z_t \\ \delta u_t \end{bmatrix}, \quad (28)$$

where the first and second order derivatives of the $Q-$function are given by

$$Q_{t,z}(\boldsymbol{z}_t, \boldsymbol{u}_t) = \left[\frac{\partial \bar{c}_t(\boldsymbol{z}_t, \boldsymbol{u}_t)}{\partial z} + A(\boldsymbol{z}_t, \boldsymbol{u}_t)^\top \mathbb{V}_{t+1,z}(\boldsymbol{z}_t, \boldsymbol{u}_t)\right],$$

$$Q_{t,u}(\boldsymbol{z}_t, \boldsymbol{u}_t) = \left[\frac{\partial \bar{c}_t(\boldsymbol{z}_t, \boldsymbol{u}_t)}{\partial u} + B(\boldsymbol{z}_t, \boldsymbol{u}_t)^\top \mathbb{V}_{t+1,z}(\boldsymbol{z}_t, \boldsymbol{u}_t)\right],$$

$$Q_{t,zz}(\boldsymbol{z}_t, \boldsymbol{u}_t) = \left[\frac{\partial^2 \bar{c}_t(\boldsymbol{z}_t, \boldsymbol{u}_t)}{\partial z^2} + A(\boldsymbol{z}_t, \boldsymbol{u}_t)^\top \mathbb{V}_{t+1,zz}(\boldsymbol{z}_t, \boldsymbol{u}_t)A(\boldsymbol{z}_t, \boldsymbol{u}_t)\right],$$

$$Q_{t,uz}(\boldsymbol{z}_t, \boldsymbol{u}_t) = \left[\frac{\partial^2 \bar{c}_t(\boldsymbol{z}_t, \boldsymbol{u}_t)}{\partial u \partial z} + B(\boldsymbol{z}_t, \boldsymbol{u}_t)^\top \mathbb{V}_{t+1,zz}(\boldsymbol{z}_t, \boldsymbol{u}_t)A(\boldsymbol{z}_t, \boldsymbol{u}_t)\right],$$

$$Q_{t,uu}(\boldsymbol{z}_t, \boldsymbol{u}_t) = \left[\frac{\partial^2 \bar{c}_t(\boldsymbol{z}_t, \boldsymbol{u}_t)}{\partial u^2} + B(\boldsymbol{z}_t, \boldsymbol{u}_t)^\top \mathbb{V}_{t+1,zz}(\boldsymbol{z}_t, \boldsymbol{u}_t)B(\boldsymbol{z}_t, \boldsymbol{u}_t)\right],$$

and the first and second order derivatives of the value functions are given by

$$\mathbb{V}_{t+1,z}(\boldsymbol{z}_t, \boldsymbol{u}_t) = \mathbb{E}_w\left[\frac{\partial V_{t+1}}{\partial z}(F(\boldsymbol{z}_t, \boldsymbol{u}_t, w))\right], \quad \mathbb{V}_{t+1,zz}(\boldsymbol{z}_t, \boldsymbol{u}_t) = \mathbb{E}_w\left[\frac{\partial^2 V_{t+1}}{\partial z^2}(F(\boldsymbol{z}_t, \boldsymbol{u}_t, w))\right].$$

Notice that the $Q$-function approximation $Q_t$ in (28) is quadratic and the matrix $\begin{bmatrix} Q_{t,zz}(\boldsymbol{z}_t, \boldsymbol{u}_t) & Q_{t,uz}(\boldsymbol{z}_t, \boldsymbol{u}_t)^\top \\ Q_{t,uz}(\boldsymbol{z}_t, \boldsymbol{u}_t) & Q_{t,uu}(\boldsymbol{z}_t, \boldsymbol{u}_t) \end{bmatrix}$ is positive semi-definite. Therefore the optimal perturbed policy $\pi_{\delta z,t}^*$ has the following closed-form solution:

$$\pi_{\delta z,t}^*(\cdot) \in \arg\min_{\delta u_t} Q_t(z_t, u_t) \implies \pi_{\delta z,t}^*(\delta z_t) = k_t(\boldsymbol{z}_t, \boldsymbol{u}_t) + K_t(\boldsymbol{z}_t, \boldsymbol{u}_t)\delta z_t, \quad (29)$$

where the controller weights are given by

$k_t(\boldsymbol{z}_t, \boldsymbol{u}_t) = -\left(Q_{t,uu}(\boldsymbol{z}_t, \boldsymbol{u}_t)\right)^{-1} Q_{t,u}(\boldsymbol{z}_t, \boldsymbol{u}_t)$ and $K_t(\boldsymbol{z}_t, \boldsymbol{u}_t) = -\left(Q_{t,uu}(\boldsymbol{z}_t, \boldsymbol{u}_t)\right)^{-1} Q_{t,uz}(\boldsymbol{z}_t, \boldsymbol{u}_t)$. Furthermore, by putting the optimal solution into the Taylor expansion of the $Q$-function $Q_t$, we get

$$Q_t(z_t, u_t) - Q_t(\boldsymbol{z}_t, \boldsymbol{u}_t) = \frac{1}{2}\begin{bmatrix} 1 \\ \delta z_t \end{bmatrix}^\top \begin{bmatrix} Q_{t,11}^*(\boldsymbol{z}_t, \boldsymbol{u}_t) & (Q_{t,21}^*(\boldsymbol{z}_t, \boldsymbol{u}_t))^\top \\ Q_{t,21}^*(\boldsymbol{z}_t, \boldsymbol{u}_t) & Q_{t,22}^*(\boldsymbol{z}_t, \boldsymbol{u}_t) \end{bmatrix} \begin{bmatrix} 1 \\ \delta z_t \end{bmatrix},$$

where the closed-loop first and second order approximations of the $Q$-function are given by

$$Q_{t,11}^*(\boldsymbol{z}_t, \boldsymbol{u}_t) = C_w^\top C_w - Q_{t,u}(\boldsymbol{z}_t, \boldsymbol{u}_t)^\top Q_{t,uu}(\boldsymbol{z}_t, \boldsymbol{u}_t)^{-1} Q_{t,u}(\boldsymbol{z}_t, \boldsymbol{u}_t),$$

$$Q_{t,21}^*(\boldsymbol{z}_t, \boldsymbol{u}_t) = Q_{t,z}(\boldsymbol{z}_t, \boldsymbol{u}_t)^\top - k_t(\boldsymbol{z}_t, \boldsymbol{u}_t)^\top Q_{t,uu}(\boldsymbol{z}_t, \boldsymbol{u}_t)K_t(\boldsymbol{z}_t, \boldsymbol{u}_t),$$

$$Q_{t,22}^*(\boldsymbol{z}_t, \boldsymbol{u}_t) = Q_{t,zz}(\boldsymbol{z}_t, \boldsymbol{u}_t) - K_t(\boldsymbol{z}_t, \boldsymbol{u}_t)^\top Q_{t,uu}(\boldsymbol{z}_t, \boldsymbol{u}_t)K_t(\boldsymbol{z}_t, \boldsymbol{u}_t).$$

Notice that at time step $t$ the optimal value function also has the following form:

$$V_t(z_t) = \min_{\delta u_t} Q_t(z_t, u_t) = \delta Q_t(\boldsymbol{z}_t, \boldsymbol{u}_t, \delta z_t, \pi_{\delta z,t}^*(\delta z_t)) + Q_t(\boldsymbol{z}_t, \boldsymbol{u}_t)$$

$$= \frac{1}{2}\begin{bmatrix} 1 \\ \delta z_t \end{bmatrix}^\top \begin{bmatrix} Q_{t,11}^*(\boldsymbol{z}_t, \boldsymbol{u}_t) & (Q_{t,21}^*(\boldsymbol{z}_t, \boldsymbol{u}_t))^\top \\ Q_{t,21}^*(\boldsymbol{z}_t, \boldsymbol{u}_t) & Q_{t,22}^*(\boldsymbol{z}_t, \boldsymbol{u}_t) \end{bmatrix} \begin{bmatrix} 1 \\ \delta z_t \end{bmatrix} + Q_t(\boldsymbol{z}_t, \boldsymbol{u}_t). \quad (30)$$

Therefore, the first and second order differential value functions can be

$$\mathbb{V}_{t,z}(\boldsymbol{z}_t, \boldsymbol{u}_t) = Q_{t,21}^*(\boldsymbol{z}_t, \boldsymbol{u}_t), \quad \mathbb{V}_{t,zz}(\boldsymbol{z}_t, \boldsymbol{u}_t) = Q_{t,22}^*(\boldsymbol{z}_t, \boldsymbol{u}_t),$$

and the value improvement at the nominal state $\boldsymbol{z}_t$ at time step $t$ is given by

$$V_t(\boldsymbol{z}_t) = \frac{1}{2}Q_{t,11}^*(\boldsymbol{z}_t, \boldsymbol{u}_t) + Q_t(\boldsymbol{z}_t, \boldsymbol{u}_t).$$

### B.3 INCORPORATING RECEDING-HORIZON TO iLQR

While iLQR provides an effective way of computing a sequence of (locally) optimal actions, it has two limitations. First, unlike RL in which an optimal Markov policy is computed, this algorithm only finds a sequence of open-loop optimal control actions under a given initial observation. Second, the iLQR algorithm requires the knowledge of a nominal (latent state and action) trajectory at every iteration, which restricts its application to cases only when real-time interactions with environment are possible. In order to extend the iLQR paradigm into the closed-loop RL setting, we utilize the concept of model predictive control (MPC) (Rawlings & Mayne, 2009; Borrelli et al., 2017) and propose the following iLQR-MPC procedure. Initially, given an initial latent state $z_0$ we generate a single nominal trajectory: $\{(\boldsymbol{z}_t, \boldsymbol{u}_t, \boldsymbol{z}_{t+1})\}_{t=0}^{T-1}$, whose sequence of actions is randomly sampled, and the latent states are updated by forward propagation of latent state dynamics (instead of interacting with environment), i.e., $\boldsymbol{z}_0 = z_0$, $\boldsymbol{z}_{t+1} \sim F(\boldsymbol{z}_t, \boldsymbol{u}_t, w_t)$, $\forall t$. Then at each time-step $k \geq 0$, starting at latent state $z_k$ we compute the optimal perturbed policy $\{\pi^*_{\delta z,t}(\cdot)\}_{t=k}^{T-1}$ using the iLQR algorithm with $T - k$ lookahead steps. Having access to the perturbed latent state $\delta z_k = z_k - \boldsymbol{z}_k$, we only deploy the first action $u^*_k = \pi^*_{\delta z,k}(\delta z_k) + \boldsymbol{u}_k$ in the environment and observe the next latent state $z_{k+1}$. Then, using the subsequent optimal perturbed policy $\{\pi^*_{\delta z,t}(\cdot)\}_{t=k+1}^{T-1}$, we generate both the estimated latent state sequence $\{\widehat{z}_t\}_{t=k+1}^{T}$ by forward propagation with initial state $\widehat{z}_{k+1} = z_{k+1}$ and action sequence $\{u^*_t\}_{t=k+1}^{T-1}$, where $u^*_t = \pi^*_{\delta z,t}(\delta \widehat{z}_t) + \boldsymbol{u}_t$, and $\delta \widehat{z}_t = \widehat{z}_t - \boldsymbol{z}_t$. Then one updates the subsequent nominal trajectory as follows: $\{(\boldsymbol{z}_t, \boldsymbol{u}_t, \boldsymbol{z}_{t+1})\}_{t=k+1}^{T-1} = \{(\widehat{z}_t, u^*_t, \widehat{z}_{t+1})\}_{t=k+1}^{T-1}$, and repeats the above procedure.

Consider the finite-horizon MDP problem $\min_{\pi_t, \forall t} \mathbb{E}\left[c_T(x_T) + \sum_{t=0}^{T-1} c_t(x_t, u_t) \mid \pi_t, P, x_0\right]$, where the optimizer $\pi$ is over the class of Markovian policies. (Notice this problem is the closed-loop version of (SOC1).) Using the above iLQR-MPC procedure, at each time step $t \in \{0, \dots, T-1\}$ one can construct a Markov policy that is in form of

$$\pi_{t,\text{iLQR-MPC}}(\cdot|x_t) := u_t^{\text{iLQR}} \text{ s.t. } \{u_t^{\text{iLQR}}, \cdots, u_{T-1}^{\text{iLQR}}\} \leftarrow \text{iLQR}(L(U, \widetilde{P}^*, \overline{c}, z_t)), \text{ with } z_t \sim E(\cdot|x_t),$$

where $\text{iLQR}(\ell_{\text{Control}}(U; \widetilde{P}^*, z))$ denotes the iLQR algorithm with initial latent state $z$. To understand the performance of this policy w.r.t. the MDP problem, we refer to the sub-optimality bound of iLQR (w.r.t. open-loop control problem in (SOC1)) in Section 3.3, as well as that for MPC policy, whose details can be found in Borrelli et al. (2017).

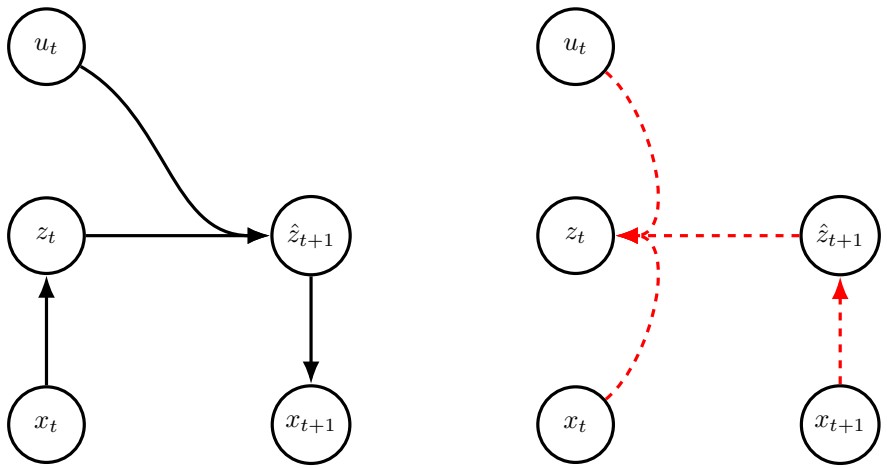

Figure 3: PCC graphical model. Left: generative links, right: recognition links.

## C  TECHNICAL PROOFS OF SECTION 4

### C.1  DERIVATION OF $R'_{3,\text{NLE-BOUND}}(\widehat{P}, Q)$ DECOMPOSITION

We derive the bound for the conditional log-likelihood $\log \widehat{P}(x_{t+1}|x_t, u_t)$.

$$
\begin{aligned}
\log \widehat{P}(x_{t+1}|x_t, u_t) &= \log \int_{z_t, \hat{z}_{t+1}} \widehat{P}(x_{t+1}, z_t, \hat{z}_{t+1}|x_t, u_t) dz_t d\hat{z}_{t+1} \\
&= \log \mathbb{E}_{Q(z_t, \hat{z}_{t+1}|x_t, x_{t+1}, u_t)} \left[ \frac{\widehat{P}(x_{t+1}, z_t, \hat{z}_{t+1}|x_t, u_t)}{Q(z_t, \hat{z}_{t+1}|x_t, x_{t+1}, u_t)} \right] \\
&\overset{(a)}{\geq} \mathbb{E}_{Q(z_t, \hat{z}_{t+1}|x_t, x_{t+1}, u_t)} \left[ \log \frac{\widehat{P}(x_{t+1}, z_t, \hat{z}_{t+1}|x_t, u_t)}{Q(z_t, \hat{z}_{t+1}|x_t, x_{t+1}, u_t)} \right] \\
&\overset{(b)}{=} \mathbb{E}_{\substack{Q(\hat{z}_{t+1}|x_{t+1}) \\ Q(z_t|\hat{z}_{t+1}, x_t, u_t)}} \left[ \log \frac{\widehat{P}(z_t|x_t)\widehat{P}(\hat{z}_{t+1}|z_t, u_t)\widehat{P}(x_{t+1}|\hat{z}_{t+1})}{Q(\hat{z}_{t+1}|x_{t+1})Q(z_t|\hat{z}_{t+1}, x_t, u_t)} \right] \\
&\overset{(c)}{=} \mathbb{E}_{Q(\hat{z}_{t+1}|x_{t+1})} \left[ \log \widehat{P}(x_{t+1}|\hat{z}_{t+1}) \right] \\
&\quad - \mathbb{E}_{Q(\hat{z}_{t+1}|x_{t+1})} \left[ D_{\text{KL}} \left( Q(z_t|\hat{z}_{t+1}, x_t, u_t) || \widehat{P}(z_t|x_t) \right) \right] \\
&\quad + H\left( Q(\hat{z}_{t+1}|x_{t+1}) \right) \\
&\quad + \mathbb{E}_{\substack{Q(\hat{z}_{t+1}|x_{t+1}) \\ Q(z_t|\hat{z}_{t+1}, x_t, u_t)}} \left[ \log \widehat{P}(\hat{z}_{t+1}|z_t, u_t) \right] \\
&= R'_{3,\text{NLE-Bound}}(\widehat{P}, Q)
\end{aligned}
$$

Where (a) holds from the log function concavity, (b) holds by the factorization $Q(z_t, \hat{z}_{t+1}|x_t, x_{t+1}, u_t) = Q(\hat{z}_{t+1}|x_{t+1})Q(z_t|\hat{z}_{t+1}, x_t, u_t)$, and (c) holds by a simple decomposition to the different components.

## C.2   Derivation of $R''_{2,\text{Bound}}(\widehat{P}, Q)$ Decomposition

We derive the bound for the consistency loss $\ell_{\text{Consistency}}(\widehat{P})$.

$$
\begin{aligned}
R''_2(\widehat{P}) &= D_{\text{KL}}\left(\widehat{P}(z_{t+1}|x_{t+1})\| \int_{z_t} \widehat{P}(\hat{z}_{t+1}|z_t, u_t)\widehat{P}(z_t|x_t)dz_t\right) \\
&\stackrel{(a)}{=} -H\left(Q(\hat{z}_{t+1}|x_{t+1})\right) - \mathbb{E}_{Q(\hat{z}_{t+1}|x_{t+1})}\left[\log \int_{z_t} \widehat{P}(\hat{z}_{t+1}|z_t, u_t)\widehat{P}(z_t|x_t)dz_t\right] \\
&= -H\left(Q(\hat{z}_{t+1}|x_{t+1})\right) - \mathbb{E}_{Q(\hat{z}_{t+1}|x_{t+1})}\left[\log \mathbb{E}_{Q(z_t|\hat{z}_{t+1},x_t,u_t)}\left[\frac{\widehat{P}(\hat{z}_{t+1}|z_t, u_t)\widehat{P}(z_t|x_t)}{Q(z_t|\hat{z}_{t+1}, x_t, u_t)}\right]\right] \\
&\stackrel{(b)}{\leq} -H\left(Q(\hat{z}_{t+1}|x_{t+1})\right) - \mathbb{E}_{\substack{Q(\hat{z}_{t+1}|x_{t+1}) \\ Q(z_t|\hat{z}_{t+1},x_t,u_t)}}\left[\log \frac{\widehat{P}(\hat{z}_{t+1}|z_t, u_t)\widehat{P}(z_t, x_t)}{Q(z_t|\hat{z}_{t+1}, x_t, u_t)}\right] \\
&\stackrel{(c)}{=} -\Bigg(-\mathbb{E}_{Q(\hat{z}_{t+1}|x_{t+1})}\left[D_{\text{KL}}\left(Q(z_t|\hat{z}_{t+1}, x_t, u_t)\|\widehat{P}(z_t|x_t)\right)\right] \\
&\qquad\qquad + H\left(Q(\hat{z}_{t+1}|x_{t+1})\right) + \mathbb{E}_{\substack{Q(\hat{z}_{t+1}|x_{t+1}) \\ Q(z_t|\hat{z}_{t+1},x_t,u_t)}}\left[\log \widehat{P}(\hat{z}_{t+1}|z_t, u_t)\right]\Bigg) \\
&= R''_{2,\text{Bound}}(\widehat{P}, Q)
\end{aligned}
$$

Where (a) holds by the assumption that $Q(\hat{z}_{t+1} \mid x_{t+1}) = \widehat{P}(z_{t+1} \mid x_{t+1})$, (b) holds from the log function concavity, and (c) holds by a simple decomposition to the different components.

# D   EXPERIMENTAL DETAILS

In the following sections we will provide the description of the data collection process, domains, and implementation details used in the experiments.

## D.1   DATA COLLECTION PROCESS

To generate our training and test sets, each consists of triples $(x_t, u_t, x_{t+1})$, we: (1) sample an underlying state $s_t$ and generate its corresponding observation $x_t$, (2) sample an action $u_t$, and (3) obtain the next state $s_{t+1}$ according to the state transition dynamics, add it a zero-mean Gaussian noise with variance $\sigma^2 I_{n_s}$, and generate it's corresponding observation $x_{t+1}$. To ensure that the observation-action data is uniformly distributed (see Section 3), we sample the state-action pair $(s_t, u_t)$ uniformly from the state-action space. To understand the robustness of each model, we consider both deterministic ($\sigma = 0$) and stochastic scenarios. In the stochastic case, we add noise to the system with different values of $\sigma$ and evaluate the models' performance under various degree of noise.

## D.2   DESCRIPTION OF THE DOMAINS

**Planar System**   In this task the main goal is to navigate an agent in a surrounded area on a 2D plane (Breivik & Fossen, 2005), whose goal is to navigate from a corner to the opposite one, while avoiding the six obstacles in this area. The system is observed through a set of $40 \times 40$ pixel images taken from the top view, which specifies the agent's location in the area. Actions are two-dimensional and specify the $x - y$ direction of the agent's movement, and given these actions the next position of the agent is generated by a deterministic underlying (unobservable) state evolution function. *Start State:* one of three corners (excluding bottom-right). *Goal State:* bottom-right corner. *Agent's Objective:* agent is within Euclidean distance of 2 from the goal state.

**Inverted Pendulum — SwingUp & Balance**   This is the classic problem of controlling an inverted pendulum (Furuta et al., 1991) from $48 \times 48$ pixel images. The goal of this task is to swing up an under-actuated pendulum from the downward resting position (pendulum hanging down) to the top position and to balance it. The underlying state $s_t$ of the system has two dimensions: angle and angular velocity, which is unobservable. The control (action) is 1-dimensional, which is the torque applied to the joint of the pendulum. To keep the Markovian property in the observation (image) space, similar to the setting in E2C and RCE, each observation $x_t$ contains two images generated from consecutive time-frames (from current time and previous time). This is because each image only shows the position of the pendulum and does not contain any information about the velocity. *Start State:* Pole is resting down (SwingUp), or randomly sampled in $\pm\pi/6$ (Balance). *Agent's Objective:* pole's angle is within $\pm\pi/6$ from an upright position.

**CartPole**   This is the visual version of the classic task of controlling a cart-pole system (Geva & Sitte, 1993). The goal in this task is to balance a pole on a moving cart, while the cart avoids hitting the left and right boundaries. The control (action) is 1-dimensional, which is the force applied to the cart. The underlying state of the system $s_t$ is 4- dimensional, which indicates the angle and angular velocity of the pole, as well as the position and velocity of the cart. Similar to the inverted pendulum, in order to maintain the Markovian property the observation $x_t$ is a stack of two $80 \times 80$ pixel images generated from consecutive time-frames. *Start State:* Pole is randomly sampled in $\pm\pi/6$. *Agent's Objective:* pole's angle is within $\pm\pi/10$ from an upright position.

**3-link Manipulator — SwingUp & Balance**   The goal in this task is to move a 3-link manipulator from the initial position (which is the downward resting position) to a final position (which is the top position) and balance it. In the 1-link case, this experiment is reduced to inverted pendulum. In the 2-link case the setup is similar to that of arcobot (Spong, 1995), except that we have torques applied to all intermediate joints, and in the 3-link case the setup is similar to that of the 3-link planar robot arm domain that was used in the E2C paper, except that the robotic arms are modeled by simple rectangular rods (instead of real images of robot arms), and our task success criterion requires both swing-up (manipulate to final position) and balance.[12] The underlying (unobservable) state $s_t$ of the system is $2N$-dimensional, which indicates the relative angle and angular velocity at each link, and the actions are $N$-dimensional, representing the force applied to each joint of the arm. The

---

[12]Unfortunately due to copyright issues, we cannot test our algorithms on the original 3-link planar robot arm domain.

state evolution is modeled by the standard Euler-Lagrange equations (Spong, 1995; Lai et al., 2015). Similar to the inverted pendulum and cartpole, in order to maintain the Markovian property, the observation state $x_t$ is a stack of two $80 \times 80$ pixel images of the $N$-link manipulator generated from consecutive time-frames. In the experiments we will evaluate the models based on the case of $N = 2$ (2-link manipulator) and $N = 3$ (3-link manipulator). *Start State:* 1st pole with angle $\pi$, 2nd pole with angle $2\pi/3$, and 3rd pole with angle $\pi/3$, where angle $\pi$ is a resting position. *Agent's Objective:* the sum of all poles' angles is within $\pm\pi/6$ from an upright position.

**TORCS Simulaotr**  This task takes place in the TORCS simulator (Wymann et al., 2000) (specifically in michegan f1 race track, only straight lane). The goal of this task is to control a car so it would remain in the middle of the lane. We restricted the task to only consider steering actions (left / right in the range of $[-1, 1]$), and applied a simple procedure to ensure the velocity of the car is always around 10. We pre-processed the observations given by the simulator ($240 \times 320$ RGB images) to receive $80 \times 80$ binary images (white pixels represent the road). In order to maintain the Markovian property, the observation state $x_t$ is a stack of two $80 \times 80$ images (where the two images are 7 frames apart - chosen so that consecutive observation would be somewhat different). The task goes as follows: the car is forced to steer strongly left (action=1), or strongly right (action=-1) for the initial 20 steps of the simulation (direction chosen randomly), which causes it to drift away from the center of the lane. Then, for the remaining horizon of the task, the car needs to recover from the drift, return to the middle of the lane, and stay there. *Start State:* 20 steps of drifting from the middle of the lane by steering strongly left, or right (chosen randomly). *Agent's Objective:* agent (car) is within Euclidean distance of 1 from the middle of the lane (full width of the lane is about 18).

### D.3 IMPLEMENTATION

In the following we describe architectures and hyper-parameters that were used for training the different algorithms.

#### D.3.1 TRAINING HYPER-PARAMETERS AND REGULIZERS

All the algorithms were trained using:

- Batch size of 128.
- ADAM (Goodfellow et al., 2016) with $\alpha = 5 \cdot 10^{-4}$, $\beta_1 = 0.9$, $\beta_2 = 0.999$, and $\epsilon = 10^{-8}$.
- $L_2$ regularization with a coefficient of $10^{-3}$.
- Additional VAE (Kingma & Welling, 2013) loss term given by $\ell_t^{\text{VAE}} = -\mathbb{E}_{q(z|x)}[\log p(x|z)] + D_{\text{KL}}(q(z|x)\|p(z))$, where $p(z) \sim \mathcal{N}(0, 1)$. The term was added with a very small coefficient of 0.01. We found this term to be important to stabilize the training process, as there is no explicit term that governs the scale of the latent space.

E2C training specifics:

- $\lambda$ from the loss term of E2C was tuned using a parameter sweep in $\{0.25, 0.5, 1\}$, and was chosen to be 0.25 across all domains, as it performed the best independently for each domain.

PCC training specifics:

- $\lambda_{\text{p}}$ was set to 1 across all domains.
- $\lambda_{\text{c}}$ was set to be 7 across all domains, after it was tuned using a parameter sweep in $\{1, 3, 7, 10\}$ on the Planar system.
- $\lambda_{\text{cur}}$ was set to be 1 across all domains without performing any tuning.
- $\{\bar{z}, \bar{u}\}$, for the curvature loss, were generated from $\{z, u\}$ by adding Gaussian noise $\mathcal{N}(0, 0.1^2)$, where $\sigma = 0.1$ was set across all domains without performing any tuning.
- Motivated by Hafner et al. (2018), we added a deterministic loss term in the form of cross entropy between the output of the generative path given the current observation and action (while taking the means of the encoder output and the dynamics model output) and the observation of the next state. This loss term was added with a coefficient of 0.3 across all domains after it was tuned using a parameter sweep over $\{0.1, 0.3, 0.5\}$ on the Planar system.

### D.3.2 NETWORK ARCHITECTURES

We next present the specific architecture choices for each domain. For fair comparison, The numbers of layers and neurons of each component were shared across all algorithms. ReLU non-linearities were used between each two layers.

**Encoder:** composed of a backbone (either a MLP or a CNN, depending on the domain) and an additional fully-connected layer that outputs mean variance vectors that induce a diagonal Gaussian distribution.

**Decoder:** composed of a backbone (either a MLP or a CNN, depending on the domain) and an additional fully-connected layer that outputs logits that induce a Bernoulli distribution.

**Dynamical model:** the path that leads from $\{z_t, u_t\}$ to $\hat{z}_{t+1}$. Composed of a MLP backbone and an additional fully-connected layer that outputs mean and variance vectors that induce a diagonal Gaussian distribution. We further added a skip connection from $z_t$ and summed it with the output of the mean vector. When using the amortized version, there are two additional outputs $A$ and $B$.

**Backwards dynamical model:** the path that leads from $\{\hat{z}_{t+1}, u_t, x_t\}$ to $z_t$. each of the inputs goes through a fully-connected layer with $\{N_z, N_u, N_x\}$ neurons, respectively. The outputs are then concatenated and pass though another fully-connected layer with $N_{\text{joint}}$ neurons, and finally with an additional fully-connected layer that outputs the mean and variance vectors that induce a diagonal Gaussian distribution.

**Planar system**

- **Input:** $40 \times 40$ images. 5000 training samples of the form $(x_t, u_t, x_{t+1})$
- **Actions space:** 2-dimensional
- **Latent space:** 2-dimensional
- **Encoder:** 3 Layers: 300 units - 300 units - 4 units (2 for mean and 2 for variance)
- **Decoder:** 3 Layers: 300 units - 300 units - 1600 units (logits)
- **Dynamics:** 3 Layers: 20 units - 20 units - 4 units
- **Backwards dynamics:** $N_z = 5, N_u = 5, N_x = 100$ - $N_{\text{joint}} = 100$ - 4 units
- **Number of control actions:** or the planning horizon $T = 40$

**Inverted Pendulum — Swing Up & Balance**

- **Input:** Two $48 \times 48$ images. 20000 training samples of the form $(x_t, u_t, x_{t+1})$
- **Actions space:** 1-dimensional
- **Latent space:** 3-dimensional
- **Encoder:** 3 Layers: 500 units - 500 units - 6 units (3 for mean and 3 for variance)
- **Decoder:** 3 Layers: 500 units - 500 units - 4608 units (logits)
- **Dynamics:** 3 Layers: 30 units - 30 units - 6 units
- **Backwards dynamics:** $N_z = 10, N_u = 10, N_x = 200$ - $N_{\text{joint}} = 200$ - 6 units
- **Number of control actions:** or the planning horizon $T = 400$

**Cart-pole Balancing**

- **Input:** Two $80 \times 80$ images. 15000 training samples of the form $(x_t, u_t, x_{t+1})$
- **Actions space:** 1-dimensional
- **Latent space:** 8-dimensional
- **Encoder:** 6 Layers: Convolutional layer: $32 \times 5 \times 5$; stride $(1, 1)$ - Convolutional layer: $32 \times 5 \times 5$; stride $(2, 2)$ - Convolutional layer: $32 \times 5 \times 5$; stride $(2, 2)$ - Convolutional layer: $10 \times 5 \times 5$; stride $(2, 2)$ - 200 units - 16 units (8 for mean and 8 for variance)
- **Decoder:** 6 Layers: 200 units - 1000 units - 100 units - Convolutional layer: $32 \times 5 \times 5$; stride $(1, 1)$ - Upsampling $(2, 2)$ - convolutional layer: $32 \times 5 \times 5$; stride $(1, 1)$ - Upsampling $(2, 2)$ - Convolutional layer: $32 \times 5 \times 5$; stride $(1, 1)$ - Upsampling $(2, 2)$ - Convolutional layer: $2 \times 5 \times 5$; stride $(1, 1)$

- **Dynamics:** 3 Layers: 40 units - 40 units - 16 units
- **Backwards dynamics:** $N_z = 10$, $N_u = 10$, $N_x = 300$ - $N_{\text{joint}} = 300$ - 16 units
- **Number of control actions:** or the planning horizon $T = 200$

### 3-link Manipulator — Swing Up & Balance

- **Input:** Two $80 \times 80$ images. 30000 training samples of the form $(x_t, u_t, x_{t+1})$
- **Actions space:** 3-dimensional
- **Latent space:** 8-dimensional
- **Encoder:** 6 Layers: Convolutional layer: $62 \times 5 \times 5$; stride $(1, 1)$ - Convolutional layer: $32 \times 5 \times 5$; stride $(2, 2)$ - Convolutional layer: $32 \times 5 \times 5$; stride $(2, 2)$ - Convolutional layer: $10 \times 5 \times 5$; stride $(2, 2)$ - 500 units - 16 units (8 for mean and 8 for variance)
- **Decoder:** 6 Layers: 500 units - 2560 units - 100 units - Convolutional layer: $32 \times 5 \times 5$; stride $(1, 1)$ - Upsampling $(2, 2)$ - convolutional layer: $32 \times 5 \times 5$; stride $(1, 1)$ - Upsampling $(2, 2)$ - Convolutional layer: $32 \times 5 \times 5$; stride $(1, 1)$ - Upsampling $(2, 2)$ - Convolutional layer: $2 \times 5 \times 5$; stride $(1, 1)$
- **Dynamics:** 3 Layers: 40 units - 40 units - 16 units
- **Backwards dynamics:** $N_z = 10$, $N_u = 10$, $N_x = 400$ - $N_{\text{joint}} = 400$ - 16 units
- **Number of control actions:** or the planning horizon $T = 400$

### TORCS

- **Input:** Two $80 \times 80$ images. 30000 training samples of the form $(x_t, u_t, x_{t+1})$
- **Actions space:** 1-dimensional
- **Latent space:** 8-dimensional
- **Encoder:** 6 Layers: Convolutional layer: $32 \times 5 \times 5$; stride $(1, 1)$ - Convolutional layer: $32 \times 5 \times 5$; stride $(2, 2)$ - Convolutional layer: $32 \times 5 \times 5$; stride $(2, 2)$ - Convolutional layer: $10 \times 5 \times 5$; stride $(2, 2)$ - 200 units - 16 units (8 for mean and 8 for variance)
- **Decoder:** 6 Layers: 200 units - 1000 units - 100 units - Convolutional layer: $32 \times 5 \times 5$; stride $(1, 1)$ - Upsampling $(2, 2)$ - convolutional layer: $32 \times 5 \times 5$; stride $(1, 1)$ - Upsampling $(2, 2)$ - Convolutional layer: $32 \times 5 \times 5$; stride $(1, 1)$ - Upsampling $(2, 2)$ - Convolutional layer: $2 \times 5 \times 5$; stride $(1, 1)$
- **Dynamics:** 3 Layers: 40 units - 40 units - 16 units
- **Backwards dynamics:** $N_z = 10$, $N_u = 10$, $N_x = 300$ - $N_{\text{joint}} = 300$ - 16 units
- **Number of control actions:** or the planning horizon $T = 200$

## E  ADDITIONAL RESULTS

### E.1  PERFORMANCE ON NOISY DYNAMICS

Table 3 shows results for the noisy cases.

Table 3: Percentage of steps in goal state. Averaged over all models (left), and over the best model (right). Subscript under the domain name is the variance of the noise that was added.

| Domain | RCE (all) | E2C (all) | PCC (all) | RCE (top 1) | E2C (top 1) | PCC (top 1) |
|---|---|---|---|---|---|---|
| $\text{Planar}_1$ | $1.2 \pm 0.6$ | $0.6 \pm 0.3$ | $\mathbf{17.9 \pm 3.1}$ | $5.5 \pm 1.2$ | $6.1 \pm 0.9$ | $\mathbf{44.7 \pm 3.6}$ |
| $\text{Planar}_2$ | $0.4 \pm 0.2$ | $1.5 \pm 0.9$ | $\mathbf{14.5 \pm 2.3}$ | $1.7 \pm 0.5$ | $15.5 \pm 2.6$ | $\mathbf{29.7 \pm 2.9}$ |
| $\text{Pendulum}_1$ | $6.4 \pm 0.3$ | $\mathbf{23.8 \pm 1.2}$ | $16.4 \pm 0.8$ | $8.1 \pm 0.4$ | $\mathbf{36.1 \pm 0.3}$ | $29.5 \pm 0.2$ |
| $\text{Cartpole}_1$ | $8.1 \pm 0.6$ | $6.6 \pm 0.4$ | $\mathbf{9.8 \pm 0.7}$ | $\mathbf{20.3 \pm 11}$ | $16.5 \pm 0.4$ | $17.9 \pm 0.8$ |
| $\text{3-link}_1$ | $0.3 \pm 0.1$ | $0 \pm 0$ | $\mathbf{0.5 \pm 0.1}$ | $1.3 \pm 0.2$ | $0 \pm 0$ | $\mathbf{1.8 \pm 0.3}$ |

### E.2 LATENT SPACE REPRESENTATION FOR THE PLANAR SYSTEM

The following figures depicts 5 instances (randomly chosen from the 10 trained models) of the learned latent space representations for both the noiseless and the noisy planar system from PCC, RCE, and E2C models.

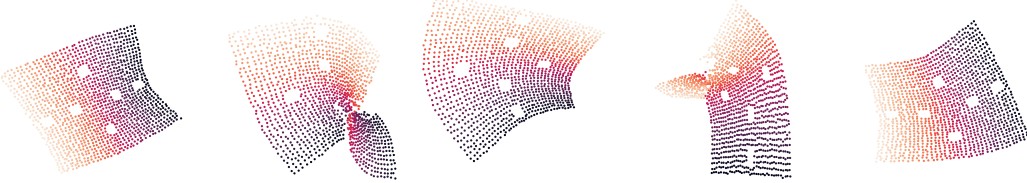

Figure 4: Noiseless Planar latent space representations using PCC.

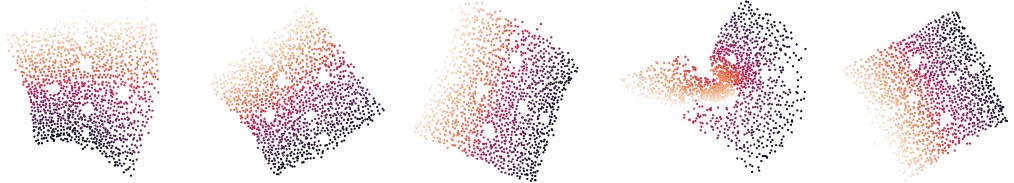

Figure 5: Noisy ($\sigma^2 = 1$) Planar latent space representations using PCC.

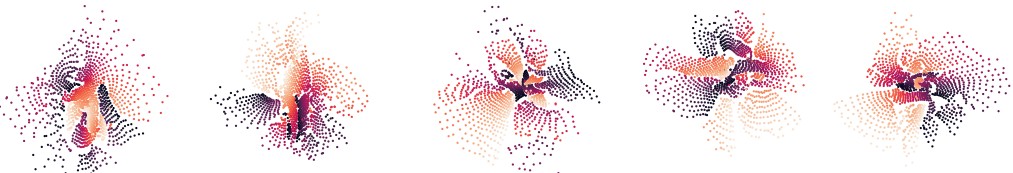

Figure 6: Noiseless Planar latent space representations using RCE.

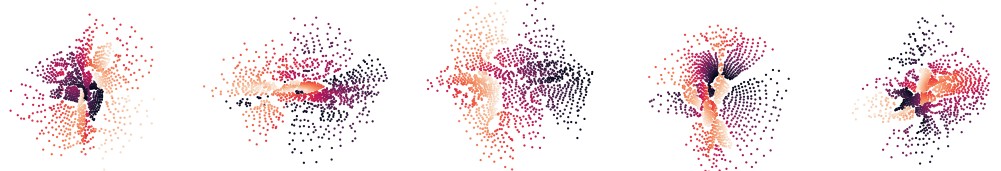

Figure 7: Noisy ($\sigma^2 = 1$) Planar latent space representations using RCE.

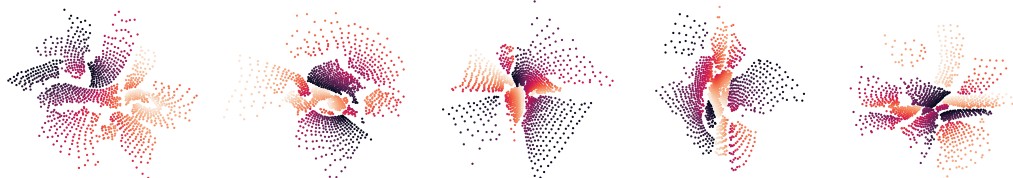

Figure 8: Noiseless Planar latent space representations using E2C.

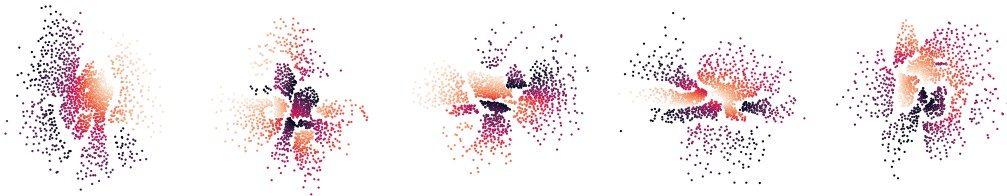

Figure 9: Noisy ($\sigma^2 = 1$) Planar latent space representations using E2C.

