# OpenReview forum: "Prediction, Consistency, Curvature: Representation Learning for Locally-Linear Control"
_ICLR.cc/2020/Conference — Accept (Poster)_

### Official Review · AnonReviewer1 · 2019-10-17
**Official Blind Review #1**

**Rating:** 6

**Review:**

This work proposes a regularization strategy for learning optimal policy for a dynamic control problem in a latent low-dimensional domain. The work is based on LCE approach, but with in-depth analysis on how to choose/design the regularization for the \hat{P} operator, which consists of an encoder, a decoder, and dynamics in the latent space. In particular, the author argued that three principles (prediction, consistency, and curvature) should be taken into consideration when designing the regularizer of the learning cost function - so that the learned latent domain can serve better for the purpose of optimizing the long-term cost in the ambient domain.

The paper is well written and pleasant to read. One possible shortcoming is that the notations are a bit dazzling. It is almost impossible to follow the notation when first reading this paper. The proofs are very lengthy and thus the reviewer did not check in detail.

The reviewer has several question:

1) Of course SOC2 makes sense. But what if one models the whole problem as an HMM, and perform control algorithms in the hidden domain of the HMM (and the hidden states can be of much smaller alphabets compared to the observable states), will there be any fundamental difference? Of course learning an HMM is challenging, but approachable. Any comments?

2) The three design principles make sense, but may need more elaboration. For example, it is a bit unclear why f_Z should be with low curvature -- does it mean that you wish the control problem in the latent domain is more like a linear dynamical system, so that the LLC algorithm works better? The argument is a bit unclear, since "locally linear" is not a rigorous term. Any smooth function is ``"locally linear". Here, how to measure the difficulty of the latent control problem may need more discussion.

Minor: btw,  (5) may contain some typos.

3) In practice, how to balance the three parameters lambda_p, lambda_c, lambda_cur?



**Experience Assessment:**

I do not know much about this area.

**Review Assessment: Checking Correctness Of Derivations And Theory:**

I did not assess the derivations or theory.

**Review Assessment: Checking Correctness Of Experiments:**

I did not assess the experiments.

**Review Assessment: Thoroughness In Paper Reading:**

I read the paper at least twice and used my best judgement in assessing the paper.

---

> ### Author Response · Authors · 2019-11-08
> **Thank you. Please find our response to your questions/comments below.**
>
> We thank the reviewer for useful comments.
>
> 1) HMM is definitely an option for learning the latent space. We note that the concepts we highlight in this paper (prediction, consistency, curvature) remain relevant even if the underlying dynamics model of choice is an HMM. However, since our paper makes the simplifying assumption that the observation space is Markovian, and since training an HMM is generally more challenging than training Markovian dynamics models, we chose not to employ an HMM in our paper.
>
> 2) The three terms in PCC are prediction, consistency, and curvature. For prediction, the goal is to enforce that the process of encoding, transitioning via the latent dynamics, and then decoding, adheres to the true observation dynamics. For consistency, the goal is to make sure that the latent dynamics is consistent with the encoded trajectory. Figure 1 clearly shows the relation/difference between the evolution of the system in the latent space and the evolution of the encoded observations. Finally, for curvature, the goal is to learn a latent space that is suitable for LLC algorithms. As stated by the reviewer, the main motivation of the curvature term is to ensure that LLC algorithms, such as iLQR, work well. When the curvature is low, the size of the neighbourhood in which the local linearity assumption holds is large, and thus, LLC algorithms work better. A more detailed discussion of the relationship between curvature and the performance of LLC algorithms can be found in Appendix A.5, in particular, see Lemma 4.
>
> 3) From Lemma 3, we see that \lambda_p and \lambda_c should be of the same order. In practice, we used this observation and optimized the hyper-parameters \lambda_p, \lambda_c, and \lambda_cur by standard grid-search.
>
> “Minor: (5) may contain some typos”
> You are right. There is a typo in (5) on the 2-norm. We will fix it in the final version of the paper.

---

### Official Review · AnonReviewer2 · 2019-10-24
**Official Blind Review #2**

**Rating:** 8

**Review:**

This paper considers from a high level the problem of learning a latent representation of high dimensional observations with underlying dynamics for control.  The authors specifically describe some desiredata for latent representations for LLC algorithms. The authors rigorously construct a learning framework that can satisfy the desiredata and then show how this can be tractably instantiated.

The paper overall is clear, however there is many equations in 4.2  with heavy subscritping making it sometimes difficult to read. The authours could attempt to better highlight the more critical parts of their propositins (e.g. eq. 8/9).

The methodology and insights appear novel and well motivated, however I am not familiar with many of the prior work.  The experiments compared to competing methods  show substantial improvement. The authors also motivate well why these improvements over the existing methods should occur and provide ablations to validate all the components of the final loss. Overall the paper appears very solid and may motivate insights and research  in more complex model based control and planning


**Experience Assessment:**

I do not know much about this area.

**Review Assessment: Checking Correctness Of Derivations And Theory:**

I did not assess the derivations or theory.

**Review Assessment: Checking Correctness Of Experiments:**

I assessed the sensibility of the experiments.

**Review Assessment: Thoroughness In Paper Reading:**

I read the paper at least twice and used my best judgement in assessing the paper.

---

> ### Author Response · Authors · 2019-11-08
> **Thank you**
>
> We thank the reviewer for appreciating our work in terms of novelty, theory, and experiments.
>
> We will improve the notations and presentation of the mathematical results, especially in Section 4.2, in the final version of the paper.

---

### Official Review · AnonReviewer4 · 2019-11-03
**Official Blind Review #4**

**Rating:** 6

**Review:**

This paper considers learning low-dimensional representations from high-dimensional observations for control purposes. The authors extend the E2C framework by introducing the new PCC-Loss function. This new loss function aims to reflect the prediction in the observation space, the consistency between latent and observation dynamics, and the low curvature in the latent dynamics. The low curvature term is used to bias the latent dynamics towards models that can be better approximated as locally linear models. The authors provide theory (error bounds) to justify their proposed PCC-Loss function. Then variational PCC is developed to make the algorithm tractable. The proposed method is evaluated in 5 different simulated tasks and compared with the original E2C method and the RCE method.
The paper is well-written.

Pros:
- The idea in this paper is quite original. The three principles used to formulate the loss function provide some new insights.

- The authors have proposed a theory to justify the use of their loss function. The technical quality of this part seems solid.

- Simulations have been used to show that the proposed PCC method outperforms E2C and RCE.

-The paper is well written.

Cons:
- The tasks in this paper are not that complicated. It is unclear whether the proposed method outperforms other model-based RL methods such as Solar and DSAE for practical robotic applications. More comparisons are needed.

- It is also not that clear why one wants to SOC3 to be close to SOC1 in the first place. It seems the true optimization problem should be posed on the space of the original state s. SOC1 is just a surrogate problem for the original problem.

- There seems to be a gap between the proposed theory  and the algorithm implementation. This makes the theory part less useful.

Overall, I think the idea in this paper is interesting. The authors have made a serious effort in coming up principles for model-based RL control. But at this moment it is not that convincing the proposed method will be the best model-based RL method for practical robotic applications. If the authors can address my comments, I will be willing to increase my score.



Minor Comments:

- It seems that for the task the authors have tested their method, it is not that difficult to directly estimate the state. Am I correct here? Can the authors make a comment on this? How to compare their approach and a more direct control approach using estimation of state s?

- I have never seen the curvature principle in any control papers. Any control reference on why this is a good principle? It seems that the linearization works well when the control inputs are around the reference points. Does the curvature really matter that much for ILQR to work?

- How to justify the Markovian assumption on x? Just by observation or there is a more principle way to test this assumption on the buffered images?

====================================================
Post-Rebuttal:
After reading the authors' response, I am changing my score to weak accept. Lemma 4 is nice. I have not seen anything similar to this in the controls literature before. The authors have addressed most of my concerns. I still have a few comments for preparing the final version of this paper.
1. I still don't see why SOC1 is the "original problem." Yes, it is assumed that the true state cannot be directly observed. But if the observations are Markov eventually, then some estimated version of the states can be obtained, right? I think treating SOC1 as the original problem is one possible way of doing things and clearly the authors have built a principled framework for doing things in this way. But treating SOC1 as the original problem seems not the only way of doing things. I hope the authors can clarify this and do not oversell the proposed approach.
2. I think it is still worth comparing SOLAR and PCC empirically. This will help the readers to choose algorithms when they need.
3. The comment on the verification of the Markov assumption is hand-waving.  The authors said "A simple test would be to see if a control algorithm with the Markovian assumption works well with our representation or not." Does this mean that the users will not be able to verify this assumption before using the proposed approach to obtain controllers? It will be helpful if the authors can explain this step for one specific example in details.


**Experience Assessment:**

I have read many papers in this area.

**Review Assessment: Checking Correctness Of Derivations And Theory:**

I assessed the sensibility of the derivations and theory.

**Review Assessment: Checking Correctness Of Experiments:**

I assessed the sensibility of the experiments.

**Review Assessment: Thoroughness In Paper Reading:**

I made a quick assessment of this paper.

---

> ### Author Response · Authors · 2019-11-08
> **Thank you. Please find our response to your questions/comments below**
>
> We thank the reviewer for the detailed and useful comments.
>
> “it is not that convincing the proposed method will be the best model-based RL method for practical robotic applications”
>
> The goal of the paper is not to come up with the best model-based RL algorithm for practical robotics applications. We aim to provide a principled framework to model the control-in-latent-space problem, and propose a concrete instantiation of the framework for learning controllable representations in particular for locally linear control (LLC) algorithms. Our method is potentially suitable for any control problem (not necessarily robotics) in which the true state space is not observed and the observations are high-dimensional. We emphasize that this problem setting of embedding high-dimensional decision processes into lower dimensional representation spaces in which classical control algorithms can be applied has been of growing interest in the recent literature (e.g. E2C, RCE, SOLAR, PlaNet), and we believe that the novel insight gained from our work (consistency, low-curvature) makes an important contribution in this space by improving significantly the robustness of existing representation for LLC approaches (E2C, RCE).
>
> PCC belongs to the family of E2C and RCE algorithms, in which learning takes place in two phases: learning the latent space and dynamics, followed by control. This is different than SOLAR in which learning the representation and control are done together in an online fashion. This is why we left comparison with SOLAR as a future work. It is important to note that for more complex problems, PCC should be implemented more interactive, by repeating its two phases several times.
>
> “why one wants to SOC3 to be close to SOC1”
> The main reason for introducing the relationship between SOC3 and SOC1 is to motivate the prediction term in the PCC model.
>
> “SOC1 is just a surrogate problem for the original problem”
> It is true that the original problem should be posed in the true state space S. However, since we assume from the very beginning that the true state of the system is not observable, SOC1 is in fact the original problem.
>
> “Gap between theory and algorithm implementation”
> The theory of the paper is mainly to motivate and support the three-part loss function used by the PCC algorithm, and not as a recipe for implementing it. Variational PCC is one possible implementation of the three-part loss. There could be many others.
>
> “directly estimating the underlying state”
> Estimating the underlying states from the observations and then conducting control in the space of the estimated states is definitely a possible approach to the problems studied in our paper. This approach requires pre-processing (state estimation) and may work well in some problems and fail in others. In this paper, we take a different approach, and instead of estimating the underlying state space, we learn a latent space suitable for the class of locally linear control (LLC) algorithms, learn the model, and conduct the control there.
>
> “curvature term and its relationship with control theory (iLQR)”
> Unlike classical control problems, in which the system dynamics is pre-defined and cannot be changed, in the embed-to-control setting, we have the luxury to enforce the latent dynamics to be suitable for our choice of control algorithm. Our main message here is that the choice of the control algorithm should play a role in learning the representation/latent space. Since the control algorithm of our choice is LLC, we believe introducing the curvature penalty term could greatly simplifies the complexity of the downstream control task. The reason behind this (intuitively) is that when the curvature is low, the size of the neighbourhood in which the local linearity assumption holds is large, and thus, LLC algorithms work better. A more detailed discussion of the relationship between curvature and the performance of LLC algorithms can be found in Appendix A.5, in particular, see Lemma 4.
>
> “Markovian observation assumption and its justification”
> Since most control algorithms work under the Markovian assumption, it is common to put the observations together in a way to be Markov or close to Markov in the observation space (the Atari domains are good examples). This is definite a pre-processing step, and it is straightforward in some problems and difficult in others. There are ways to test whether our representation is (approximately) Markov or not. Having knowledge about the physics of the system is definitely helpful. A simple test would be to see if a control algorithm with the Markovian assumption works well with our representation or not.

---

### Decision · Program_Chairs · 2019-12-19

**Decision:**

Accept (Poster)

**Comment:**

This paper studies optimal control with low-dimensional representation.  The paper presents interesting progress, although I urge the authors to address all issues raised by reviewers in their revisions.